# Synthesis of New Azetidine and Oxetane Amino Acid Derivatives through Aza-Michael Addition of NH-Heterocycles with Methyl 2-(Azetidin- or Oxetan-3-Ylidene)Acetates

**DOI:** 10.3390/molecules28031091

**Published:** 2023-01-21

**Authors:** Emilis Gudelis, Sonata Krikštolaitytė, Monika Stančiauskaitė, Urtė Šachlevičiūtė, Aurimas Bieliauskas, Vaida Milišiūnaitė, Rokas Jankauskas, Neringa Kleizienė, Frank A. Sløk, Algirdas Šačkus

**Affiliations:** 1Institute of Synthetic Chemistry, Kaunas University of Technology, K. Baršausko g. 59, LT-51423 Kaunas, Lithuania; 2Department of Organic Chemistry, Kaunas University of Technology, Radvilėnų pl. 19, LT-50254 Kaunas, Lithuania; 3Vipergen ApS, Gammel Kongevej 23A, V DK-1610 Copenhagen, Denmark

**Keywords:** azetidine, oxetane, heterocyclic amines, heterocyclic amino acids, aza-Michael addition, Suzuki–Miyaura cross-coupling reaction

## Abstract

In this paper, a simple and efficient synthetic route for the preparation of new heterocyclic amino acid derivatives containing azetidine and oxetane rings was described. The starting (*N*-Boc-azetidin-3-ylidene)acetate was obtained from (*N*-Boc)azetidin-3-one by the DBU-catalysed Horner–Wadsworth–Emmons reaction, followed by aza-Michael addition with NH-heterocycles to yield the target functionalised 3-substituted 3-(acetoxymethyl)azetidines. Methyl 2-(oxetan-3-ylidene)acetate was obtained in a similar manner, which was further treated with various (*N*-Boc-cycloaminyl)amines to yield the target 3-substituted 3-(acetoxymethyl)oxetane compounds. The synthesis and diversification of novel heterocyclic amino acid derivatives were achieved through the Suzuki–Miyaura cross-coupling from the corresponding brominated pyrazole–azetidine hybrid with boronic acids. The structures of the novel heterocyclic compounds were confirmed via ^1^H-, ^13^C-, ^15^N-, and ^19^F-NMR spectroscopy, as well as HRMS investigations.

## 1. Introduction

In heterocyclic chemistry, four-membered saturated heterocycles containing one nitrogen or oxygen atom are known as azetidines and oxetanes, respectively [1,2]. The pharmacophore subunit of azetidine in aza-heterocyclic molecules is used for a wide variety of natural and synthetic products exhibiting a variety of biological activities [3,4]. For example, the azetidine subunit is a structure derived from some alkaloids from marine sources, which show relatively potent cytotoxic activity against tumour cells as well as antibacterial activity [5,6]. An azetidine ring is also present in the molecular structure of the well-known antihypertensive drug azelnidipine, which is a dihydropyridine calcium channel blocker [7,8].

Azetidine carboxylic acids are important scaffolds and building blocks for obtaining various biologically active heterocyclic compounds and peptides [9,10,11]. Specifically, *L*-azetidine-2-carboxylic acid is found in nature in sugar beets (*Beta vulgaris*) and is a gametocidal agent [12]. In addition, this amino acid is an inhibitor of collagen synthesis that is antiangiogenic [13]. Azetidine-2-carboxylic acid (**I**) and its 3-aryl derivatives, which are *L*-proline analogues, have also been widely used as building blocks to prepare small peptides (Figure 1) [14,15]. Additionally, both azetidine-3-carboxylic (**II**) and 3-(4-oxymethylphenyl)azetidine-3-carboxylic (**III**) acids, which are conformationally constrained analogues of β-proline, were employed for the preparation of endomorphin tetrapeptides **IV** and **V**, respectively [16,17]. Recently, He and Hartwig developed a simple and efficient method for 3-aryl- and 3-heteroarylazetidine-3-carboxylic acid compounds via a Pd*-*catalysed cross-coupling between *t*-butyl (*N*-benzylazetidine-3-yl) carboxylate and (het)aryl halides [18]. Such azetidine derivatives, including compound **VI**, can act as analogues of a pain medication named Meperidine **VII** [19].

(Azetidin-3-yl)acetic acid **VIII** could be used as a structural analogue for 4-aminobutanoic acid (GABA) [9]. (3-Arylazetidin-3-yl)acetates **IX** and **X** are used for the preparation of pharmaceutically active agents, including the positive allosteric modulators of GABA_A_ receptors [20]. Chalyk et al. developed a general method for isoxazole-containing building blocks, namely azetidine amino ester **XI** as a 5-aminopentanoic acid (*δ*-aminovaleric acid) ester analogue [21]. 5-Aminopentanoic acid is a naturally occurring amino acid and a methylene homologue of GABA [22]. Recently, we developed efficient protocols that provide easy access to highly functional heterocyclic compounds by combining heterocyclic moieties with both carboxylic ester functional groups and cycloaminyl units, such as the δ-amino esters azetidine derivatives **XII** and **XIII** [23,24].

The pharmacophoric subunit of oxetane, containing various organic compounds, has been extensively studied in medicinal chemistry [25]. This oxetane ring structure is widespread in natural products and has been found to exhibit a number of biological activities. Oxetin, i.e., 3-amino-2-oxetanecarboxylic acid **XIV**, was isolated from the broth of *Streptomyces* and has been shown to possess antibacterial and herbicidal effects (Figure 2) [26]. The oxetane subunit is a structure derived from natural or synthetic taxanes clinically used in cancer chemotherapy [27]. Notably, the oxetane nucleoside of antibiotic oxetanocin A, isolated from natural sources, inhibits the replication of the human immunodeficiency virus (HIV) [28]. 3-Aminooxetane-3-carboxylic acid **XV**, a structural analogue for glycine, was reported as a modulator of the *N*-methyl-D-aspartate (NMDA) receptor complex [29].

Carreira et al. investigated various properties of oxetanes as substituents, leading to many useful developments, especially in the use of oxetanes as substitutes for carbonyl groups, which is of considerable interest due to their similar dipoles and H-bonding ability [30,31]. Powell et al. reported the preparation of derivatives in which the central C=O amide bond of a tripeptide was replaced by the oxetane nucleus [32]. Several reports are devoted to the synthesis and evaluation of the physicochemical and metabolic properties of δ-amino acid oxetane derivatives, such as compound **XVI** [33].

This study aimed to develop and synthesise new heterocyclic amino acid derivatives containing azetidine and oxetane rings. Such amino acid compounds offer valuable properties as isosteres, new conformationally restricted amino acids, and building blocks that can be used as potentially biologically active substances and peptides, as well as for the generation of DNA*-*encoded peptide libraries [34].

## 2. Results and Discussion

The strategy for the synthesis of novel heterocyclic amino acids containing azetidine rings is outlined in Figure 1. The synthetic sequence began with methyl (*N*-Boc-azetidin-3-ylidene)acetate **3**, prepared from azetidin-3-one **2** through the Horner–Wadsworth–Emmons (HWE) reaction. The HWE reaction is one of the most reliable and common synthetic methods for preparing substituted alkene products from aldehydes and ketones with phosphonate esters [35]. Yang et al. recently developed a simple method for the preparation of compound **3** from methyl 2-(dimethoxyphosphoryl)acetate **1** with a 60% suspension of NaH in mineral oil in dry THF, followed by the addition of azetidin-2-one **2**. The reaction was then quenched with water, and the resulting aqueous solution was extracted with EtOAc and concentrated in vacuo; finally, the residue was purified via flash column chromatography [36]. We carried out a similar synthesis for **3**, but this method differed in that the corresponding residue was purified through two-stage vacuum distillation in a Büchi oven (kugelrohr) [37] at a reduced pressure of 4 × 10^–3^ bar by first distilling the volatile fraction at 90 °C for some time (usually approximately 1 h) and then changing the collection vessel and increasing the temperature to 130 °C to produce the pure product **3** (yield 72%). This method for the preparation of compound **3** allows the purification of large quantities and works well while trying to avoid stubborn impurities such as mineral oil.

Next, having obtained α,β−unsaturated ester **3**, aza-Michael addition was carried out with heterocyclic aliphatic and heterocyclic aromatic amines for the formation of heterocyclic amino acid blocks **4**. Aza-Michael addition is a powerful and versatile method for constructing C–N bonds containing various highly functional organic compounds, which has remained an important challenge over the last decade [38,39]. In particular, this synthetic strategy has been applied for the preparation of NH-heterocyclic derivatives, such as azetidine, pyrrolidine, piperidine, and morpholine-saturated heterocycles [40], as well as 1*H*-pyrazole [41], 1*H*-imidazole [42], 1*H*-1,2,4-triazole [43], 1*H*-indole [44], 1*H*-indazole [45], 1*H*-benzotriazole [46], and related aromatic heterocycles. The latter compounds are widely used as important pharmacophores for pharmaceutical development [47,48]. Various methods for the aza-Michael addition reaction have been developed using a variety of promoters, such as inorganic and organic bases, proton acids, Lewis acids, and enzymes [49,50,51,52]. Aza-Michael addition greatly benefits from its mild reaction conditions, and the choice of a non-nucleophilic base 1,8-diazabicyclo[5.4.0]undec-7-ene (DBU) helps to prevent side reactions, for example, the cleavage of the ester group, which can be caused by other strong bases such as hydroxides [53,54,55]. Yeom et al. developed a convenient method for the preparation of functionalised derivatives from cyclic amines with methyl acrylate through aza-Michael reaction using a sub-stoichiometric amount of DBU as an effective promoter [56]. Xu et al. reported a reaction of 4-(7-SEM-pyrrolo[2,3-*d*]pyrimidin-4-yl)pyrazole with a 2-(1-(ethylsulfonyl)azetidin-3-ylidene)acetonitrile, heated to 60 °C in acetonitrile-containing DBU, thus yielding a baricitinib heterocyclic intermediate [57].

Methyl (*N*-Boc-azetidin-3-ylidene)acetate **3** was reacted (Figure 1) with azetidine and DBU in the solvent acetonitrile at 65 °C for 4 h to obtain 1,3′-biazetidine **4a** with a 64% yield. Compound **4a** was subjected to a detailed spectral analysis. Absorption bands characteristic of the esters at 1731 (C=O, ester) and 1694 (C=O, Boc) cm^−1^ were observed on the IR spectrum of compound **4a**. In the ^1^H-NMR spectrum of compound **4a**, four characteristic methylene protons CH_2_-2,4 were observed in the regions of δ 3.69–3.86 and 3.94–4.06 ppm, which appeared significantly broadened due to the conformational dynamics of the 3,3-substituted azetidine moiety in the solvent. The second azetidine ring, containing the symmetric fragment CH_2_CH_2_CH_2_, showed methylene protons CH_2_-2′,4′ appearing as a triplet at δ 3.29 (^3^*J* = 7.2 Hz) ppm, while two protons CH_2_-3′ appeared as a pentet at δ 2.05 (^3^*J* = 7.2 Hz) ppm. The ^1^H-^15^N HMBC spectrum of **4a** showed the characteristic resonances of the nitrogen atoms of the azetidine rings at δ −315.4 (N-1, Boc-azetidine) and −337.8 ppm (N-1′, azetidine), respectively. 3-Hydroxy-1,3′-biazetidine **4b** was synthesised by analogy to **4a** from 3-hydroxyazetidine by aza-Michael addition with a 62% yield. The key information for structure elucidation was also obtained from the ^1^H-^15^N HMBC spectrum. As expected, the ^15^N chemical shifts of the N-1 Boc-azetidine (δ −315.0 ppm) and N-1′ azetidine (δ −350.2 ppm) atoms were highly comparable to those of compound **4a**. The observed chemical shifts of the azetidine derivatives were consistent with the data reported in the literature [23,24,58,59].

The reaction of **3** with pyrrolidine under these conditions resulted in compound **4c** with a 61% yield, while the obtained 3,3-difluoropyrrolidine led to compound **4d** with a 64% yield. Although the basicity of 3,3-difluoropyrrolidine was significantly lower (pKa 7.5) than that of pyrrolidine (pKa 11.3), it did not affect the reaction in any way [60]. 1-(Azetidin-3-yl)piperidine **4e** was isolated from reaction **3** with piperidine with a 75% yield. 3-(4-Hydroxypiperidin-1-yl)azetidines **4f** and **4g** were formed from either 4-hydroxypiperidine or 4-hydroxy-4-phenylpiperidine through aza-Michael addition with 75% and 66% yields, respectively. ^1^H-^15^N HMBC experiments for the products **4f** and **4g** confirmed the proposed structures of the isomeric piperidines, as the Boc-azetidine-ring protons H-2 and H-4 showed interactions with the nitrogens N-1′ of the piperidine rings at δ –324.2 ppm and –324.8 ppm, respectively. When the 2,3-unsaturated ester **3** was used with morpholine, adduct **4h** was obtained with a 73% yield after 4 h. It was observed that increasing the size of the heterocyclic aliphatic amines from a four- to a six-membered ring system did not adversely affect the reaction, and all adducts were obtained in moderate-to-good yields. A similar reaction was carried out with 2,3-dihydro-1*H*-isoindoline, which generated compound **4i** with a 64% yield.

Furthermore, we studied the reaction of 2,3–unsaturated ester **3** with heterocyclic aromatic amines (Figure 1). Baricitinib, a disease-modifying antirheumatic drug*,* contains a 3-(pyrazol-1-yl)azetidine skeleton [57]. The aza-Michael addition of 1*H*-pyrazole, 4-bromo-1*H*-pyrazole, and 3-trifluoromethyl-1*H*-pyrazole was carried out under the same conditions as above (DBU and solvent acetonitrile), but its duration was longer (16 h) than with heterocyclic aliphatic amines (4 h); consequently, the 3-(pyrazol-1-yl)azetidine adducts **4j**, **4k**, and **4l** reached 83%, 82%, and 73% yields, respectively. It is worth noting that brominated pyrazoles are useful synthetic intermediates in the search for biologically active compounds that are capable of undergoing transition metal-catalysed cross-coupling reactions [61,62]. The trifluoromethyl group is present in many pharmacologically active molecules, including fluorinated pyrazoles [63,64].

In the ^1^H-NMR spectrum of the 3-(pyrazol-1-yl)azetidine derivative **4j**, the methylene protons from the azetidine moiety appeared as two doublets resonating at δ 4.28 and 4.42 ppm (^2^*J*_Ha,Hb_ = 9.6 Hz), while the aromatic pyrazole protons showed three signals at δ 6.29–6.30 (m, 4′-H), 7.54 (d, *J* = 1.4 Hz, 3′-H), and 7.63 (d, *J* = 2.4 Hz, 5′-H) ppm. The NOEs were exhibited between the pyrazole-ring proton 5′-H and the azetidine 2(4)-H_a_ protons. The ^13^C-NMR spectrum confirmed the pyrazole moiety’s carbon signals, appearing at δ 106.0 (C-4′), 127.8 (C-5′), and 140.0 (C-3′) ppm. The ^1^H-^15^N HMBC spectrum of **4j** exhibited the characteristic resonances of nitrogen atoms at δ –316.6 (azetidine N-1), –163.5 (pyrazole N-1′), and –81.2 (pyrazole N-2′) ppm.

The reaction of 2,3-unsaturated ester **3** with 3-(3-trifluoromethyl)-1*H*-pyrazole could yield regioisomers **4l** and **A**, but only compound **4l** was obtained (Figure 3). The regiochemistry of compound **4l** was confirmed with a NOESY experiment, which exhibited NOEs between the pyrazole proton 5′-H and the azetidine 2(4)-H_a_ protons. In the case of compound **A**, it would not be possible to have NOEs between the protons of the pyrazole and azetidine moieties. In addition, the ^1^H-^13^C HMBC spectrum of the molecule **4l** showed a long-range correlation between the 5′-H pyrazole proton (δ 7.72 ppm) and the quaternary carbon of azetidine C-3 at δ 57.8 ppm, as well as a three-bond correlation with the quaternary carbon of pyrazole C-3′ at δ 143.0 (q, ^2^*J*_CF_ = 38.4 Hz) ppm [65].

In principle, the use of indazole as an aza-Michael donor can result in the formation of two additional products, N-1 and N-2 adducts, because of its tautomerism. However, Jiang et al. successfully developed an efficient method for the synthesis of the desired 1-substituted 1*H*-indazole compound with a 52% yield through the direct aza-Michael addition of indazole to an α,β-unsaturated malonate compound using Cs_2_CO_3_ as a catalyst [66]. Recently, Yang et al. reported a synthetic approach for synthesising 1-substituted 1*H*-indazoles via the DBU-catalysed aza-Michael reaction of 1*H*-indazole with enones. This reaction produced regioselective compounds with good substrate tolerance, mild reaction conditions, and high-to-excellent yields (up to 93%) [45].

We investigated the aza-Michael reaction of indazole with methyl (*N*-Boc-azetidine-3-ylidene)acetate **3** for possible regioisomers. The reaction was monitored via LC/MS, and the full conversion of the starting materials was observed after 16 h. The reaction of the starting materials in DBU in the solvent acetonitrile at 65 °C led to regioisomer **4m** as the sole product with a moderate 69% isolated yield. The unambiguous formation of **4m** was easily deduced from ^1^H-^15^N HMBC spectral data, as it clearly showed a strong three-bond correlation between the indazole nitrogen N-1′ (δ –190.8 ppm) with indazole 3′-H (δ 7.99 ppm) and 7′-H (δ 7.39 ppm) protons and azetidine methylene CH_2_-2,4 (δ 4.76 ppm) protons, correspondingly. The regiochemistry of compound **4m** was confirmed with a NOESY experiment, which exhibited NOEs between the pyrazole proton 7′-H and the azetidine 2(4)-H_a_ protons (Figure 3).

The aza-Michael addition reactions of 1*H*-imidazole, 1*H*-benzimidazole, and 1*H*-indole with (*N*-Boc-azetidin-3-ylidene)acetate **3** were also applied to produce azetidine-imidazole **4n**, azetidine-benzimidazole **4o,** and azetidine-indole **4p** heterocyclic compounds with 53%, 56%, and 55% yields, respectively. The structures of the newly synthesised heterocyclic compounds **4n**–**p** were described and confirmed via NMR spectroscopy (Appendix A).

Furthermore, we investigated the coupling of triazole aromatic amines, 1,2,4-triazole and 1,2,3-benzotriazole, with precursor **3** (Figure 2). In the case of the unsubstituted 1,2,4-triazole, two tautomeric forms containing an NH moiety are possible. Bulger et al. reported the alkylation of 1,2,4-triazole with alkyl halides and DBU as a base and THF as a solvent, which afforded alkylated N-1 and N-4 isomers with a consistent regioselectivity of about 90:10 [67]. However, Behn′s group reported a DBU- or alkali-salt-catalysed aza-Michael reaction of 1,2,4-triazole with α,β-unsaturated ketones, regioselectively producing only 1-substituted 1,2,4-triazoles as the N-1 adducts [68].

In the present work, the treatment of 1,2,4-triazole with α,β-unsaturated ester **3** was carried out in acetonitrile in the presence of DBU to generate compound **4q** as a single product but with a low percentage yield of 46% (Table 1, Entry 1). Therefore, the reaction conditions were optimised. It was found that the target product **4q** was not formed in the absence of a catalyst (Entry 2). Some salts such as LiF and LiCl did not affect the reaction (Entries 3,4). However, inorganic bases such as Cs_2_CO_3_, KOAc and K_3_PO_4_ produced **4q** with a moderate yield (Entries 5–7). The highest yield of **4q** was obtained in the presence of K_2_CO_3_ in acetonitrile (Entry 8). Additionally, the effect of the solvents on the reaction was evaluated (Entries 9–11). Ethanol proved to be less effective on the assessed reaction with a final yield of only 44%. Although 1,4-dioxane achieved a 60% product yield and proved to be efficient, MeCN was still better, reaching a 65% yield for the same reaction.

The regiochemistry of compound **4q** was confirmed with a NOESY experiment, which exhibited NOEs between the azetidine protons 2(4)-H_a_ at δ 4.43 ppm and the 1,2,4-triazole proton 5-H at δ 8.33 ppm. The ^1^H-^15^N HMBC experiment revealed the corresponding three-bond connectivities of azetidine methylene CH_2_-2,4 and acetate methylene CH_2_ protons with the 1,2,4-triazole nitrogen N-1 (pyrrole-type) at δ –156.5 ppm (Figure 4a).

The aza-Michael reaction of benzotriazole with methyl acrylate was reported to form N-1 and N-2 adducts in a mixture of benzotriazol-1-yl-propionic and benzotriazol-2-yl-propionic acid methyl esters by using anhydrous potassium phosphate (K_3_PO_4_) as a catalyst [54], while the 1,4-conjugated aza-Michael addition of benzotriazole to dienones catalysed by potassium acetate (KOAc) yielded only the corresponding N-1 isomer [56]. Recently, Chen et al. reported an efficient, regio- and enantioselective aza-Michael reaction for the synthesis of the N-1 isomers from benzotriazole with α-substituted β-nitroacrylates catalysed by a chiral organocatalyst [49].

In our work, the reaction of 1,2,3-benzotriazole with α,β-unsaturated ester **3** in acetonitrile in the presence of K_2_CO_3_ led to the formation of a mixture of regioisomers **4r** and **4s** in a ratio of approximately 4:3, with a total yield of 76%. Discrimination between the regioisomeric N-1 and N-2 adducts **4r** and **4s** was based on the data from ^1^H-^15^N HMBC and ^1^H-^1^H NOESY experiments. The unambiguous formation of regioisomer **4r** was easily deduced with a NOESY experiment, which exhibited NOEs between the 1*H*-1,2,3-benzotriazole-ring proton 7-H and the azetidine 2(4)-H_a_ protons. The ^1^H-^15^N HMBC experiment on the asymmetric 1*H*-1,2,3-benzotriazole fragment of compound **4r** revealed the chemical shifts of the pyrrole-type nitrogen N-1 (δ –148.6 ppm) and the pyridine-type nitrogens N-2 and N-3 (δ –7.0 and –42.0 ppm, respectively) (Figure 4b). The second regioisomer, **4s**, containing a symmetrical *2H*-1,2,3-benzotriazole fragment, was easily assigned from appropriate correlations in the ^1^H-^15^N HMBC spectrum between the equivalent H-4 and H-7 aromatic protons (δ 7.40 ppm) and equivalent N-1 and N-3 nitrogen atoms (δ –69.1 ppm) (Figure 4c).

One of the most effective methods for the structural diversification of aromatic and heterocyclic building blocks is functionalisation through Pd-catalysed Suzuki–Miyaura cross-coupling reactions [69]. With compound **4k** containing a 4-bromopyrazole moiety in hand, we further investigated Pd-catalysed coupling with organoboronic acids (Figure 3). Several coupling systems were evaluated, namely Pd(dba)_2_–K_3_PO_4_ in DCM [70], Pd(OAc)_2_–Cs_2_CO_3_ in EtOH/H_2_O [71], Pd(PPh_3_)_4_–K_2_CO_3_ in toluene/MeOH [72], and Pd(PPh_3_)_4_–K_3_PO_4_ in 1,4-dioxane [73]. The best Suzuki–Miyaura cross-coupling result was achieved using Pd(PPh_3_)_4_ as a catalyst, using K_3_PO_4_ as a base, and performing the reaction at 100 °C in 1,4-dioxane [24]. Under these conditions, the target product **5a** was obtained with an excellent (94%) yield. Compounds **5b**–**f** were obtained from the aforementioned compound **4k** with methylphenyl- and methoxyphenyl-boronic acids through Pd-catalysed coupling with the same reagents under analogous conditions as those used for compound **5a**. Target products **5b**–**e** were achieved with moderate yields of 70–80%, while the target product **5f** was obtained with a low yield (29%). Methoxyphenylboronic acids with a substituent in the ortho position reacted less efficiently than those with a substituent in the meta and/or para position. Target compounds **5g** and **5h**, containing fluoro and chloro moieties, were obtained with fair yields of 63% and 53%, respectively. Compound **4k** was reacted with pyridinyl- and thienylboronic acid, generating products **5i**–**m** with 37–64% yields.

Next, we explored the Suzuki–Miyaura cross-coupling reaction with cyclopropylboronic acid (Figure 3). The cyclopropyl group is an increasingly common structural motif in pharmaceutically active molecules [74]. The synthesis of compound **5n** was carried out with cyclopropylboronic acid under the same reaction conditions as described above, using the Pd(PPh_3_)_4_–K_3_PO_4_ system and refluxing the reaction in 1,4-dioxane, but the desired product was not obtained. Wallace et al. demonstrated a synthetic approach where aryl bromides reacted with cyclopropylboronic acid in a Pd(PPh_3_)_4_–K_3_PO_4_ system in toluene [75]. Changing the solvent to toluene had a significant effect on the reaction and, finally, the target product **5n** was obtained, but only with a 31% yield; however, the cross-coupling reaction in the P(cHex)_3_–Pd(OAc)_2_–K_3_PO_4_ system in toluene afforded compound **5n** with a sufficient (64%) yield [76].

The structures of synthesised compounds **5a**–**n** were confirmed using NMR spectroscopic methods. For example, in the ^1^H-NMR spectrum of compound **5a**, two proton singlets from the pyrazole ring were exhibited at δ 7.81 (5-H) and 7.74 (3-H) ppm, while five protons of the phenyl ring appeared at δ 7.17–7.41 ppm. The azetidine-ring signals of the diastereotopic methylene protons were observed as two doublets at δ 4.24 and 4.40 ppm (^2^*J*_Ha,Hb_ = 9.6 Hz). In the ^1^H-NMR spectrum of compounds **5b** and **5c**, the characteristic methyl protons from the methylphenyl moiety appeared in the regions of δ 2.28 and 2.31 ppm, respectively, while the methyl protons from the methoxyphenyl moiety in compounds **5d**–**5f** appeared in the region of δ 3.75–3.84 ppm. The ^1^H-NMR spectroscopic data for the methoxy-substituted derivatives **5d**–**5f** showed an evident effect of this group on the chemical shift of the aromatic protons [77]. For example, in the case of compound **5d**, the aromatic protons (3-H, 5-H) located at the ortho position to the 4-methoxy group were observed upfield (δ 6.83 ppm, d, *J* = 8.8 Hz), whereas the protons located at the meta position (2-H, 6-H) were observed downfield (δ 7.32 ppm, d, *J* = 8.8 Hz). ^15^N-NMR spectroscopic data for the methoxy-substituted derivative **5k** showed a significant effect on the ^15^N chemical shift of the pyridin-3-yl moiety (δ –116.0 ppm), which was greatly shifted upfield compared with **5j** (δ –69.5 ppm).

The ^1^H-NMR spectrum of compound **5n** showed characteristic resonances for the cyclopropyl moiety, where the methylene protons appeared as multiplets at δ 0.41–0.45 and 0.74–0.78 ppm, and the methine proton appeared as a multiplet at δ 1.58–1.62 ppm. A comparison between the DEPT-90, DEPT-135, and ^13^C-NMR spectra of compound **5n** clearly indicated the characteristic signals of the cyclopropyl-ring skeleton carbons, namely the methine carbon C-1′ (δ 5.3 ppm) and methylene carbons C-2′ and C-3′ (δ 7.6 ppm) [78].

Following the successful completion of the aza-Michael addition reactions resulting in amino acid-like blocks containing an azetidine core, we explored another four-membered heterocycle: oxetane. The target oxetane compounds were synthesised as depicted in Figure 4. The starting oxetan-3-one **6** was used in the Horner–Wadsworth–Emmons reaction with methyl-2-(dimethoxyphosphoryl)acetate **1** to obtain methyl (oxetan-3-ylidene)acetate **7** with a 73% yield according to a procedure similar to that described in the patent literature [79]. With compound **7** in our hands, compound **8a** was easily synthesised from 3-*N*-Boc-aminoazetidine hydrochloride with a 71% yield. The reaction was carried out at 45 °C for 24 h in acetonitrile in the presence of DBU.

The structure of compound **8a** was confirmed through spectral investigations. The IR spectrum of compound **8a** revealed a N–H stretching vibration band at 3314 cm^−1^ and a C=O stretching vibration band at 1719 cm^−1^ (CH_2_C=O and Boc groups). The ^1^H-NMR spectrum of compound **8a** showed characteristic resonance for the Boc-group methyl protons as a singlet at δ 1.44 ppm, and the ester CH_3_O protons appeared as a singlet that overlapped with the protons of the azetidine moiety at δ 3.64–3.71 ppm. The signals of the oxetane ring for the diastereotopic protons of both methylene groups (CH_2_-2,4) were observed as two doublets at δ 4.57 and 4.71 ppm (^2^*J*_Ha,Hb_ = 7.2 Hz). Both methylene protons (C′H_2_-2,4) of the azetidine ring showed signals in the form of broadened multiplets in the region of δ 3.11–3.24 and 3.64–3.71 ppm, and the methine proton (C′H-3) appeared as a multiplet in the region of δ 4.10–4.35 ppm. A relatively broad peak of the NH proton was observed at δ 4.98–5.06 ppm. The ^1^H-^15^N HSQC experiment indicated that the aforementioned proton had one-bond connectivity with the nitrogen NH-Boc at δ –288.5 ppm, while the oxetane-ring protons showed a ^1^H-^15^N HMBC correlation with azetidine nitrogen at δ –348.4 ppm. The ^13^C-NMR spectrum of **8a** revealed the characteristic signals of the oxetane-ring skeleton carbons at δ 76.0 (C-2,4) and 61.7 (C-3) ppm, while the azetidine-ring skeleton carbons resonated at δ 55.7 (C′-2,4) and 40.7 (C′-3) ppm.

In addition, compounds **8b**–**e** were obtained from 2,3-unsaturated ester **7** with saturated chiral cyclic amines—namely, (*S*)- and (*R*)-3-(Boc-amino)pyrrolidines, and (*S*)- and (*R*)-3-(Boc-amino)piperidines. The synthesised compounds **8b**–**e** exhibited optical activity, and the corresponding (*S*)- or (*R*)-enantiomers rotated the plane of plane-polarised light in opposite directions.

Finally, the aza-Michael addition of methyl (oxetan-3-ylidene)acetate **7** with 4-(Boc-amino)- and 4-(Boc-aminomethyl)piperidines was carried out under the same conditions as above, and adducts **8f** and **8g** reached 58% and 55% yields, respectively. The structures of the newly synthesised oxetane derivatives **8b**–**g** were described and confirmed via NMR spectroscopy (Appendix A).

## 3. Materials and Methods

### 3.1. General Information

All starting materials were purchased from commercial suppliers and were used as received. Flash column chromatography was performed on Silica Gel 60 Å (Merck KGaA, Darmstadt, Germany). Vacuum distillation was performed in a Büchi Model B580 GKR oven (Büchi Labortechnik AG, Flawil, Switzerland). Thin-layer chromatography was carried out on Silica Gel plates (Merck Kieselgel 60 F254) and visualised by UV light (254 nm) (Merck KGaA, Darmstadt, Germany). Melting points were determined using a Büchi M-565 melting point apparatus (Büchi Labortechnik AG, Flawil, Switzerland) and were uncorrected. The IR spectra were recorded on a Bruker Vertex 70v FT-IR spectrometer (Bruker Optik GmbH, Ettlingen, Germany) using neat samples and are reported in the frequency of absorption (cm^–1^). Mass spectra were obtained using a Shimadzu LCMS-2020 (ESI+) spectrometer (Shimadzu Corporation, Kyoto, Japan). High-resolution mass spectra were measured using a Bruker MicrOTOF-Q III (ESI+) apparatus (Bruker Daltonik GmbH, Bremen, Germany). Accurate measurements were achieved using the internal mass calibration of each sample using sodium formate calibration solution as a standard procedure, with a standard deviation always less than 1 ppm [80]. In addition, all data files were recalibrated with an internal standard of sodium formate injected prior to initial sample elution for each sample. Optical rotation data were recorded on a UniPol L SCHMIDT+HAENSCH polarimeter (concentration of compound (g/100 mL) and were included in calculations automatically (Windaus-Labortechnik GmbH & Co. KG, Clausthal-Zellerfeld, Germany). ^1^H-NMR and ^13^C-NMR spectra were recorded from CDCl_3_ solutions at 25 °C on a Bruker Avance III 400 instrument (400 MHz for ^1^H, 100 MHz for ^13^C) using a directly detecting BBO probe (Bruker BioSpinInternational *AG*, Faellanden, Switzerland) and a Bruker Avance III 700 instrument (700 MHz for ^1^H, 176 MHz for ^13^C) equipped with a 5 mm TCI ^1^H-^13^C/^15^N/D z-gradient cryoprobe (Bruker BioSpin GmbH, Rheinstetten, Germany). ^19^F-NMR spectra (376.46 MHz, absolute referencing via *Ξ* ratio) were obtained on a Bruker Avance III 400 instrument with a ′directly′ detecting broadband observe probe (BBO). ^15^N-NMR spectra were recorded from CDCl_3_ solutions at 25 °C on either a Bruker Avance III 400 instrument (40 MHz for ^15^N) using a directly detecting BBO probe or on a Bruker Avance III 700 instrument (71 MHz for ^15^N). The chemical shifts (δ), expressed in ppm, were relative to tetramethylsilane (TMS). ^15^N-NMR spectra were referenced against neat external nitromethane (coaxial capillary). The following abbreviations were used in reporting the NMR data: Az, azetidine; Cpr, cyclopropane; *i*-Ind, *iso*-Indoline; Morph, morpholine; Ox, oxetane; Pip, piperidine; Ph, phenyl; Pyr, pyridine; Pyrr, pyrrolidine; Prz, pyrazole; Idz, indazole; Imid, imidazole; Bim, benzimidazole; Ind, indole; Btz, benzotriazole; Trz, triazole; Thio, thiophene.

### 3.2. Synthetic Procedures

#### 3.2.1. *tert*-Butyl 3-(2-methoxy-2-oxoethylidene)azetidine-1-carboxylate (**3**)

Neat methyl 2-(dimethoxyphosphoryl)acetate **1** (13.8 g, 76 mmol) was added to a suspension of NaH (60% dispersion in mineral oil) (3.12 g, 78 mmol) in dry THF (250 mL). After 30 min, a solution of 1-Boc-3-azetidinone **2** (13.0 g, 76 mmol) in dry THF (50 mL) was added, and the resulting mixture was stirred for 1 h. The reaction was quenched by the addition of water (250 mL). The organic layer was separated, and the aqueous one was extracted with ethyl acetate (3 × 150 mL). The combined organic solutions were dried over anhydr. Na_2_SO_4_, and then the solvents were removed under reduced pressure. Purification was conducted at 130 °C and 4 × 10^−3^ bar pressure via distillation in vacuo to give **3** (12.44 g, 72%) as a colorless oil. IR (ν_max_, cm^−1^): 2968, 1720 (C=O), 1701 (C=O). ^1^H NMR (700 MHz, CDCl_3_): δ_H_ ppm 1.39 (s, 9H, C(CH_3_)_3_), 3.66 (s, 3H, OCH_3_), 4.52–4.54 (m, 2H, Az 2,4-H_b_), 4.74–4.76 (m, 2H, Az 2,4-H_a_), 5.72–5.73 (m, 1H, CHCO). ^13^C NMR (176 MHz, CDCl_3_): δ_C_ ppm 28.3 (C(CH_3_)_3_), 51.5 (COOCH_3_), 57.9 and 60.3 (Az C-2,4), 80.1 (C(CH_3_)_3_), 113.3 (CHCOOCH_3_), 153.1 (Az C-3), 156.2 (COOC(CH_3_)_3_), 165.7 (COOCH_3_). The HRMS (ESI^+^) for C_11_H_17_NNaO_4_ ([M+Na]^+^) was calcd. as 250.1051 and found to be 250.1050.

#### 3.2.2. General Procedure for Compounds **4a**–**p**

An appropriate *N*-heterocyclic compound (5.2 mmol), DBU (0.79 g, 5.2 mmol), and *tert*-butyl 3-(2-methoxy-2-oxoethylidene)azetidine-1-carboxylate **3** (1.18 g, 5.2 mmol) were dissolved in acetonitrile (3.6 mL) and stirred at 65 °C for 4–16 h. The reaction was quenched by the addition of water (15 mL) and extracted with ethyl acetate (3 × 10 mL). The combined organic solutions were dried over anhydr. Na_2_SO_4_, and then the solvents were removed under reduced pressure. Purification was conducted via flash chromatography.

##### *tert*-Butyl 3′-(2-methoxy-2-oxoethyl)[1,3′-biazetidine]-1′-carboxylate (**4a**)

The sample was prepared from **3** (1.18 g, 5.2 mmol), azetidine hydrochloride (0.49 g, 5.2 mmol), and DBU (0.79 g, 5.2 mmol). The reaction was time 4 h. The obtained residue was purified by flash chromatography (eluent *n*-hexane/ethyl acetate, *v*/*v*, 4:1) to give **4a** (0.95 g, 64%) as a white solid, mp 79.3–80.4 °C (ethyl acetate). IR (ν_max_, cm^−1^): 1731 (C=O), 1694 (C=O). ^1^H-NMR (700 MHz, CDCl_3_): δ_H_ ppm 1.45 (s, 9H, C(CH_3_)_3_), 2.05 (p, *J* = 7.2 Hz, 2H, Az′ 3-H), 2.55 (s, 2H, CH_2_CO), 3.29 (t, *J* = 7.2 Hz, 4H, Az′ 2,4-H), 3.69 (s, 3H, COOCH_3_), 3.69–3.86 (m, 2H, Az 2,4-H_b_), 3.94–4.06 (m, 2H, Az 2,4-H_a_). ^13^C-NMR (176 MHz, CDCl_3_): δ_C_ ppm 16.0 (Az′ C-3), 28.4 (C(CH_3_)_3_), 40.5 (CH_2_COOCH_3_), 48.2 (Az′ C-2,4), 51.7 (COOCH_3_), 52.6–54.1 (Az C-2,4), 57.1 (Az C-3), 79.6 (C(CH_3_)_3_), 156.3 (COOC(CH_3_)_3_), 171.0 (COOCH_3_). ^15^N-NMR (71 MHz, CDCl_3_): δ_N_ ppm –337.8 (Az), –315.4 (Boc-Az). The HRMS (ESI^+^) for C_14_H_25_N_2_O_4_ ([M+H]^+^) was calcd. as 285.1809 and found to be 285.1809.

##### *tert*-Butyl 3-hydroxy-3′-(2-methoxy-2-oxoethyl)[1,3′-biazetidine]-1′-carboxylate (**4b**)

The sample was prepared from **3** (1.18 g, 5.2 mmol), 3-hydroxyazetidine hydrochloride (0.57 g, 5.2 mmol), and DBU (0.79 g, 5.2 mmol). The reaction time 4 h. The obtained residue was purified by flash chromatography (eluent *n*-hexane/ethyl acetate, *v*/*v*, 4:1) to give **4b** (0.97 g, 62%) as a colorless oil. IR (ν_max_, cm^−1^): 3406 (O-H), 1738 (C=O), 1698 (C=O). ^1^H-NMR (700 MHz, CDCl_3_): δ_H_ ppm 1.45 (s, 9H, C(CH_3_)_3_), 2.58 (s, 2H, CH_2_CO), 3.08–3.15 (m, 2H, Az′ 2,4-H), 3.41–3.48 (m, 1H, OH), 3.58–3.61 (m, 2H, Az′ 2,4-H) 3.68 (s, 3H, COOCH_3_), 3.74–3.89 (m, 2H, Az 2,4-H_b_), 3.93–4.05 (m, 2H, Az 2,4-H_a_), 4.43 (p, *J* = 5.9 Hz, 1H, Az′ 3-H). ^13^C-NMR (176 MHz, CDCl_3_): δ_C_ ppm 28.4 (C(CH_3_)_3_), 40.7 (CH_2_COOCH_3_), 51.7 (COOCH_3_), 53.0–54.5 (Az C-2,4), 56.9 (Az C-3), 57.7 (Az′ C-2,4), 61.1 (Az′ C-3), 79.9 (C(CH_3_)_3_), 156.3 (COOC(CH_3_)_3_), 170.9 (COOCH_3_). ^15^N-NMR (71 MHz, CDCl_3_): δ_N_ ppm –350.2 (Az), –315.0 (Boc-Az). The HRMS (ESI^+^) for C_14_H_24_N_2_NaO_5_ ([M+Na]^+^) was calcd. as 323.1577 and found to be 323.1580.

##### *tert*-Butyl 3-(2-methoxy-2-oxoethyl)-3-(pyrrolidin-1-yl)azetidine-1-carboxylate (**4c**)

The sample was prepared from **3** (1.18 g, 5.2 mmol), pyrrolidine (0.37 g, 5.2 mmol), and DBU (0.79 g, 5.2 mmol). The reaction time was 4 h. The obtained residue was purified by flash chromatography (eluent *n*-hexane/ethyl acetate, *v*/*v*, 4:1) to give **4c** (0.95 g, 61%) as a yellowish oil. IR (ν_max_, cm^−1^): 1737 (C=O), 1698 (C=O). ^1^H-NMR (700 MHz, CDCl_3_): δ_H_ ppm 1.37 (s, 9H, C(CH_3_)_3_), 1.69–1.73 (m, 4H, Pyrr 2 × CH_2_), 2.58–2.64 (m, 4H, Pyrr 2 × CH_2_), 2.67 (s, 2H, CH_2_CO), 3.61 (s, 3H, OCH_3_), 3.71–3.84 (m, 2H, Az 2,4-H_b_), 3.94 (d, *J* = 9.2 Hz, 2H, Az 2,4-H_a_). ^13^C-NMR (176 MHz, CDCl_3_): δ_C_ ppm 24.0 (Pyrr 2 × CH_2_), 28.4 (C(CH_3_)_3_), 41.3 (CH_2_COOCH_3_), 46.7 (Pyrr 2 × CH_2_), 51.7 (COOCH_3_), 54.9–56.6 (Az C-2,4), 56.7 (Az C-3), 79.5 (C(CH_3_)_3_), 156.3 (COOC(CH_3_)_3_), 171.4 (COOCH_3_). The HRMS (ESI^+^) for C_15_H_27_N_2_O_4_ ([M+H]^+^) was calcd. as 299.1965 and found to be 299.1965.

##### *tert*-Butyl 3-(3,3-difluoropyrrolidin-1-yl)-3-(2-methoxy-2-oxoethyl)azetidine-1-carboxylate (**4d**)

The sample was prepared from **3** (1.18 g, 5.2 mmol), 3,3-difluoropyrrolidine (0.56 g, 5.2 mmol), and DBU (0.79 g, 5.2 mmol). The reaction time was 4 h. The obtained residue was purified by flash chromatography (eluent *n*-hexane/ethyl acetate, *v*/*v*, 4:1) to give **4d** (1.11 g, 64%) as a yellowish oil. IR (ν_max_, cm^−1^): 1738 (C=O), 1698 (C=O), 1392, 1367, 1147, 1112. ^1^H-NMR (700 MHz, CDCl_3_): δ_H_ ppm 1.45 (s, 9H, C(CH_3_)_3_), 2.26 (tt, *J* = 14.3, 6.8 Hz, 2H, Pyrr CH_2_), 2.73 (s, 2H, CH_2_CO), 2.91 (t, *J* = 6.9 Hz, 2H, Pyrr CH_2_), 3.09 (t, *J* = 13.7 Hz, 2H, Pyrr CH_2_), 3.69 (s, 3H, OCH_3_), 3.85–3.91 (m, 2H, Az 2,4-H_b_), 3.98 (d, *J* = 9.5 Hz, 2H, Az 2,4-H_a_). ^13^C-NMR (176 MHz, CDCl_3_): δ_C_ ppm 28.3 (C(CH_3_)_3_), 35.2 (t, *J* = 24.1 Hz) (Pyrr C-4), 40.9 (CH_2_COOCH_3_), 45.2 (t, *J* = 3.6 Hz) (Pyrr C-5), 51.8 (COOCH_3_), 55.4 (t, *J* = 30.1 Hz) (Pyrr C-2), 54.5–55.8 (Az C-2,4), 56.4 (Az C-3), 79.9 (C(CH_3_)_3_), 129.1 (t, *J* = 247.8 Hz) (Pyrr C-3), 156.2 (COOC(CH_3_)_3_), 170.7 (COOCH_3_). ^19^F-NMR (376 MHz, CDCl_3_): δ_F_ ppm from –93.8 to –94.1 (m). The HRMS (ESI^+^) for C_15_H_25_F_2_N_2_O_4_ ([M+H]^+^) was calcd. as 335.1777 and found to be 335.1777.

##### *tert*-Butyl 3-(2-methoxy-2-oxoethyl)-3-(piperidin-1-yl)azetidine-1-carboxylate (**4e**)

The sample was prepared from **3** (1.18 g, 5.2 mmol), piperidine (0.44 g, 5.2 mmol), and DBU (0.79 g, 5.2 mmol). The reaction time was 4 h. The obtained residue was purified by flash chromatography (eluent *n*-hexane/ethyl acetate, *v*/*v*, 4:1) to give **4e** (1.22 g, 75%) as a colorless oil. IR (ν_max_, cm^−1^): 1733 (C=O), 1696 (C=O). ^1^H-NMR (400 MHz, CDCl_3_): δ_H_ ppm 1.32–1.39 (m, 11H, C(CH_3_)_3_, Pip CH_2_), 1.45–1.51 (m, 4H, Pip 2 × CH_2_), 2.27–2.29 (m, 4H, Pip 2 × CH_2_), 2.54 (s, 2H, CH_2_CO), 3.61 (s, 3H, COOCH_3_), 3.68 (d, *J* = 8.4 Hz, 2H, Az 2,4-H_b_), 3.78–3.88 (m, 2H, Az 2,4-H_a_). ^13^C-NMR (101 MHz, CDCl_3_) δ_C_ ppm 24.4 (Pip CH_2_), 26.2 (Pip 2 × CH_2_), 28.4 (C(CH_3_)_3_), 34.8 (CH_2_COOCH_3_), 46.5 (Pip 2 × CH_2_), 51.8 (COOCH_3_), 57.9 (Az C-3), 57.3–58.2 (Az C-2,4), 79.3 (C(CH_3_)_3_), 156.4 (COOC(CH_3_)_3_), 172.2 (COOCH_3_). The HRMS (ESI^+^) for C_16_H_29_N_2_O_4_ ([M+H]^+^) was calcd. as 313.2121 and found to be 313.2121.

##### *tert*-Butyl 3-(4-hydroxypiperidin-1-yl)-3-(2-methoxy-2-oxoethyl)azetidine-1-carboxylate (**4f**)

The sample was prepared from **3** (1.18 g, 5.2 mmol), 4-hydroxypiperidine (0.53 g, 5.2 mmol), and DBU (0.79 g, 5.2 mmol). The reaction time was 6 h. The obtained residue was purified by flash chromatography (eluent *n*-hexane/ethyl acetate, *v*/*v*, 2:1) to give **4f** (1.28 g, 75%) as a slightly yellow solid, mp 71.4–72.9 °C (ethyl acetate). IR (ν_max_, cm^−1^): 3429 (O-H), 1731 (C=O), 1680 (C=O). ^1^H-NMR (700 MHz, CDCl_3_): δ_H_ ppm 1.44 (s, 9H, C(CH_3_)_3_), 1.55 (dtd, *J* = 12.7, 9.0, 2.8 Hz, 2H, Pip CH_2_), 1.87–1.90 (m, 2H, Pip CH_2_), 2.22 (ddd, *J* = 11.9, 9.2, 3.0 Hz, 2H, Pip CH_2_), 2.63–2.66 (m, 4H, Pip CH_2_, CH_2_CO), 3.68 (s, 3H, OCH_3_), 3.78 (d, *J* = 8.5 Hz, 2H, Az 2,4-H_b_), 3.86–4.98 (m, 2H, Az 2,4-H_a_). ^13^C-NMR (176 MHz, CDCl_3_): δ_C_ ppm 28.4 (C(CH_3_)_3_), 34.6 (Pip 2 × CH_2_), 35.1 (Pip 2 × CH_2_), 43.0 (CH_2_COOCH_3_), 51.9 (COOCH_3_), 57.6 (Az C-3), 57.2–58.3 (Az C-2,4), 67.6 (Pip CHOH), 79.5 (C(CH_3_)_3_), 156.4 (COOC(CH_3_)_3_), 172.0 (COOCH_3_). ^15^N-NMR (71 MHz, CDCl_3_): δ_N_ –324.2 (Pip), –317.0 (Az). The HRMS (ESI^+^) for C_16_H_29_N_2_O_5_ ([M+H]^+^) was calcd. as 329.2071 and found to be 329.2071.

##### *tert*-Butyl 3-(4-hydroxy-4-phenylpiperidin-1-yl)-3-(2-methoxy-2-oxoethyl)azetidine-1-carboxylate (**4g**)

The sample was prepared from **3** (1.18 g, 5.2 mmol), 4-hydroxy-4-phenylpiperidine (0.92 g, 5.2 mmol), and DBU (0.79 g, 5.2 mmol). The reaction time was 6 h. The obtained residue was purified by flash chromatography (eluent *n*-hexane/ethyl acetate, *v*/*v*, 2:1) to give **4g** (1.39 g, 66%) as a white solid, mp 143.1–144.2 °C (ethyl acetate). IR (ν max, cm^−1^): 3391 (O-H), 1728 (C=O), 1672 (C=O). ^1^H-NMR (700 MHz, CDCl_3_): δ_H_ ppm 1.44 (s, 9H, C(CH_3_)_3_), 1.76 (d, *J* = 12.7 Hz, 2H, Pip CH_2_), 2.09 (t, *J* = 10.5 Hz, 2H, Pip CH_2_), 2.59–2.65 (m, 4H, Pip 2 × CH_2_), 2.70 (s, 2H, CH_2_CO), 3.70 (s, 3H, OCH_3_), 3.83 (d, *J* = 8.5 Hz, 2H, Az 2,4-H_b_), 3.89–4.02 (m, 2H, Az 2,4-H_a_), 7.26–7.28 (m, 1H, Ph CH), 7.36 (t, *J* = 7.7 Hz, 2H, Ph 2 × CH), 7.47–7.52 (m, 2H, Ph 2 × CH). ^13^C-NMR (176 MHz, CDCl_3_): δ_C_ ppm 28.4 (C(CH_3_)_3_), 35.0 (Pip 2 × CH_2_), 38.8 (Pip 2 × CH_2_), 41.7 (CH_2_COOCH_3_), 51.9 (COOCH_3_), 57.8 (Az C-3), 57.2–58.2 (Az C-2,4), 71.3 (Pip CH_2_COH(Ph)), 79.5 (C(CH_3_)_3_), 124.5 (Ph 2 × CH), 127.1 (Ph CH), 128.4 (Ph 2 × CH), 148.2 (Ph C), 156.4 (COOC(CH_3_)_3_), 172.1 (COOCH_3_). ^15^N-NMR (71 MHz, CDCl_3_): δ_N_ ppm –324.8 (Pip), –316.9 (Az). The HRMS (ESI^+^) for C_22_H_33_N_2_O_5_ ([M+H]^+^) was calcd. as 405.2384 and found to be 405.2384.

##### *tert*-Butyl 3-(2-methoxy-2-oxoethyl)-3-(morpholin-4-yl)azetidine-1-carboxylate (**4h**)

The sample was prepared from **3** (1.18 g, 5.2 mmol), morpholine (0.45 g, 5.2 mmol), and DBU (0.79 g, 5.2 mmol). The reaction time was 6 h. The obtained residue was purified by flash chromatography (eluent *n*-hexane/ethyl acetate, *v*/*v*, 2:1) to give **4h** (1.19 g, 73%) as a clear oil. IR (ν_max_, cm^−1^): 1733 (C=O), 1697 (C=O). ^1^H-NMR (700 MHz, CDCl_3_): δ_H_ ppm 1.37 (s, 9H, C(CH_3_)_3_), 2.38–2.45 (m, 4H, Morph 2 × CH_2_), 2.60 (s, 2H, CH_2_CO), 3.57–3.67 (m, 7H, OCH_3_, Morph 2 × CH_2_), 3.75–3.90 (m, 4H, Az 2,4-H). ^13^C-NMR (176 MHz, CDCl3): δ_C_ ppm 28.3 (C(CH_3_)_3_), 35.8 (CH_2_COOCH_3_), 46.0 (Morph 2 × CH_2_), 51.8 (COOCH_3_), 56.2–57.3 (Az C-2,4), 57.5 (Az C-3), 67.2 (Morph 2 × CH_2_), 79.6 (C(CH_3_)_3_), 156.4 (COOC(CH_3_)_3_), 171.6 (COOCH_3_). The HRMS (ESI^+^) for C_15_H_26_N_2_NaO_5_ ([M+Na]^+^) was calcd. as 337.1734 and found to be 337.1734.

##### *tert*-Butyl 3-(1,3-dihydro-2*H*-isoindol-2-yl)-3-(2-methoxy-2-oxoethyl)azetidine-1-carboxylate (**4i**)

The sample was prepared from **3** (1.18 g, 5.2 mmol), isoindoline hydrochloride (0.53 g, 5.2 mmol), and DBU (0.79 g, 5.2 mmol). The reaction time was 6 h. The obtained residue was purified by flash chromatography (eluent *n*-hexane/ethyl acetate, *v*/*v*, 2:1) to give **4i** (1.15 g, 64%) as a white solid, mp 102.1–102.9 °C (ethyl acetate). The yield was 64%. IR (ν_max_, cm^−1^): 1729 (C=O), 1692 (C=O). ^1^H-NMR (700 MHz, CDCl_3_): δ_H_ ppm 1.45 (s, 9H, C(CH_3_)_3_), 2.85 (s, 2H, CH_2_CO), 3.67 (s, 3H, CH_3_), 3.95–4.03 (m, 2H, Az 2,4-H), 4.14–4.21 (m, 6H, Az 2,4-H, *i*-Ind 2 × CH_2_), 7.18–7.23 (m, 4H, *i*-Ind 4 × CH). ^13^C-NMR (176 MHz, CDCl_3_): δ_C_ ppm 28.4 (C(CH_3_)_3_), 42.2 (CH_2_COOCH_3_), 51.8 (COOCH_3_), 53.2 (*i*-Ind 2 × CH_2_), 55.2–56.5 (Az C-2,4), 57.2 (Az C-3), 79.7 (C(CH_3_)_3_), 122.5 (*i*-Ind 2 × CH), 127.0 (*i*-Ind 2 × CH), 138.9 (*i*-Ind 2 × CH), 156.2 (COOC(CH_3_)_3_), 171.0 (COOCH_3_). The HRMS (ESI^+^) for C_19_H_27_N_2_O_4_ ([M+H]^+^) was calcd. as 347.1965 and found to be 347.1965.

##### *tert*-Butyl 3-(2-methoxy-2-oxoethyl)-3-(1*H*-pyrazol-1-yl)azetidine-1-carboxylate (**4j**)

The sample was prepared from **3** (1.18 g, 5.2 mmol), pyrazole (0.35 g, 5.2 mmol), and DBU (0.79 g, 5.2 mmol). The reaction time was 16 h. The obtained residue was purified by flash chromatography (eluent *n*-hexane/ethyl acetate, *v*/*v*, 4:1) to give **4j** (1.27 g, 83%) as a colorless oil. IR (ν_max_, cm^−1^): 1731 (C=O), 1692 (C=O). ^1^H-NMR (700 MHz, CDCl_3_): δ_H_ ppm 1.45 (s, 9H, C(CH_3_)_3_), 3.23 (s, 2H, CH_2_CO), 3.61 (s, 3H, COOCH_3_), 4.28 (d, *J* = 9.6 Hz, 2H, Az 2,4-H_b_), 4.42 (d, *J* = 9.6 Hz, 2H, Az 2,4-H_a_), 6.29–6.30 (m, 1H, Prz CH), 7.54 (d, *J* = 1.4 Hz, 1H, Prz CH), 7.63 (d, *J* = 2.4 Hz, 1H, Prz CH). ^13^C-NMR (176 MHz, CDCl_3_): δ_C_ ppm 28.3 (C(CH_3_)_3_), 42.2 (CH_2_COOCH_3_), 51.9 (COOCH_3_), 56.9 (Az C-3), 59.8 (Az C-2,4), 80.3 (C(CH_3_)_3_), 106.0 (Prz CH), 127.8 (Prz CH), 140.0 (Prz CH), 156.1 (COOC(CH_3_)_3_), 169.9 (COOCH_3_). ^15^N-NMR (71 MHz, CDCl_3_): δ_N_ ppm –316.6 (Az), –163.5 (Prz N-2), –81.2 (Prz N-1). The HRMS (ESI^+^) for C_14_H_21_N_3_NaO_4_ ([M+Na]^+^) was calcd. as 318.1424 and found to be 386.1424.

##### *tert*-Butyl 3-(4-bromo-1*H*-pyrazol-1-yl)-3-(2-methoxy-2-oxoethyl)azetidine-1-carboxylate (**4k**)

The sample was prepared from **3** (1.18 g, 5.2 mmol), 4-bromopyrazole (0.76 g, 5.2 mmol), and DBU (0.79 g, 5.2 mmol). The reaction time was 16 h. The obtained residue was purified by flash chromatography (eluent *n*-hexane/ethyl acetate, *v*/*v*, 4:1) to give **4k** (1.60 g, 82%) as a white solid, mp 98.8–100.1 °C (ethyl acetate). IR (ν_max_, cm^−1^): 1729 (C=O), 1690 (C=O). ^1^H-NMR (700 MHz, CDCl_3_): δ_H_ ppm 1.45 (s, 9H, C(CH_3_)_3_), 3.21 (s, 2H, CH_2_CO), 3.64 (s, 3H, OCH_3_), 4.26 (d, *J* = 9.7 Hz, 2H, Az 2,4-H_b_), 4.38 (d, *J* = 9.7 Hz, 2H, Az 2,4-H_a_), 7.49 (s, 1H, Prz CH), 7.68 (s, 1H, Prz CH). ^13^C-NMR (176 MHz, CDCl_3_): δ_C_ ppm 28.3 (C(CH_3_)_3_), 41.8 (CH_2_COOCH_3_), 52.0 (COOCH_3_), 57.6 (Az C-3), 59.8 (Az C-2,4), 80.4 (C(CH_3_)_3_), 93.7 (Prz CH), 128.2 (Prz CH), 140.5 (Prz C), 156.0 (COOC(CH_3_)_3_), 169.6 (COOCH_3_). ^15^N-NMR (71 MHz, CDCl_3_): δ_N_ ppm –317.2 (Az), –162.5 (Prz N-1), –77.9 (Prz N-2). The HRMS (ESI^+^) for C_14_H_20_BrN_3_NaO_4_ ([M+Na]^+^) was calcd. as 396.0530 and found to be 396.0529.

##### *tert*-Butyl 3-(2-methoxy-2-oxoethyl)-3-[3-(trifluoromethyl)-1*H*-pyrazol-1-yl]azetidine-1-carboxylate (**4l**)

The sample was prepared from **3** (1.18 g, 5.2 mmol), 3-(trifluoromethyl)pyrazole (0.71 g, 5.2 mmol), and DBU (0.79 g, 5.2 mmol). The reaction time was 16 h. The obtained residue was purified by flash chromatography (eluent *n*-hexane/ethyl acetate, *v*/*v*, 4:1) to give **4l** (1.38 g, 73%) as a white solid, mp 91.6–92.9 °C (ethyl acetate). IR (ν_max_, cm^−1^): 1738 (C=O), 1693 (C=O), 1155 (C-F), 1117 (C-F). ^1^H-NMR (700 MHz, CDCl_3_): δ_H_ ppm 1.45 (s, 9H, C(CH_3_)_3_), 3.24 (s, 2H, CH_2_CO), 3.63 (s, 3H, OCH_3_), 4.30 (d, *J* = 9.8 Hz, 2H, Az 2,4-H_b_), 4.44 (d, *J* = 9.7 Hz, 2H, Az 2,4-H_a_), 6.55 (d, *J* = 2.4 Hz, 1H, Prz CH), 7.72 (d, *J* = 2.3 Hz, 1H, Prz CH). ^13^C-NMR (176 MHz, CDCl_3_): δ_C_ ppm 28.3 (C(CH_3_)_3_), 42.1 (CH_2_COOCH_3_), 52.0 (COOCH_3_), 57.8 (Az C-3), 59.6 (Az C-2,4), 80.6 (C(CH_3_)_3_), 104.6 (q, ^3^*J*_C,F_ = 2.3 Hz, Prz C-4), 121.0 (q, ^1^*J*_C,F_ = 268.6 Hz, CF_3_), 129.7 (Prz C-5), 143.0 (q, ^2^*J*_C,F_ = 38.4 Hz, Prz C-3), 156.0 (COOC(CH_3_)_3_), 169.7 (COOCH_3_). ^15^N-NMR (71 MHz, CDCl_3_): δ_N_ ppm –317.2 (Az), –159.9 (Prz N-2), –79.8 (Prz N-1). ^19^F-NMR (376 MHz, CDCl_3_): δ_F_ ppm –62.1 (s, CF_3_). The HRMS (ESI^+^) for C_15_H_20_F_3_N_3_NaO_4_ ([M+Na]^+^) was calcd. as 386.1298 and found to be 386.1298.

##### *tert*-Butyl 3-(1*H*-indazol-1-yl)-3-(2-methoxy-2-oxoethyl)azetidine-1-carboxylate (**4m**)

The sample was prepared from **3** (1.18 g, 5.2 mmol), indazole (0.61 g, 5.2 mmol), and DBU (0.79 g, 5.2 mmol). The reaction time was 16 h. The obtained residue was purified by flash chromatography (eluent *n*-hexane/acetone, *v*/*v*, 4:1) to give **4m** (1.24 g, 69%) as a yellowish oil. IR (ν_max_, cm^−1^): 1738 (C=O), 1699 (C=O). ^1^H-NMR (700 MHz, CDCl_3_): δ_H_ ppm 1.47 (s, 9H, C(CH_3_)_3_), 3.20 (s, 2H, CH_2_CO), 3.50 (s, 3H, COOCH_3_), 4.47 (d, *J* = 9.3 Hz, 2H, Az 2,4-H_b_), 4.76 (d, *J* = 9.2 Hz, 2H, Az 2,4-H_a_), 7.15–7.21 (m, 1H, Idz 5-H), 7.39 (m, 2H, Idz 6-H and 7-H), 7.74 (d, *J* = 8.0 Hz, 1H, Idz 4-H), 7.99 (s, 1H, Idz 3-H). ^13^C-NMR (176 MHz, CDCl_3_): δ_C_ ppm 28.3 (C(CH_3_)_3_), 42.4 (CH_2_COOCH_3_), 51.9 (COOCH_3_), 56.9 (Az C-3), 59.0 (Az C-2,4), 80.3 (C(CH_3_)_3_), 109.9 (Idz C-7), 121.1 (Idz C-5), 121.6 (Idz C-4), 125.1 (Idz C-3a), 126.7 (Idz C-6), 133.8 (Idz C-3), 138.2 (Idz C-7a), 156.3 (COOC(CH_3_)_3_), 169.8 (COOCH_3_). ^15^N-NMR (71 MHz, CDCl_3_): δ_N_ ppm –316.3 (Az), –190.8 (Idz N-1), –67.7 (Idz N-2). The HRMS (ESI^+^) for C_18_H_23_N_3_NaO_4_ ([M+Na]^+^) was calcd. 368.1581 and found to be 368.1585.

##### *tert*-Butyl 3-(1*H*-imidazol-1-yl)-3-(2-methoxy-2-oxoethyl)azetidine-1-carboxylate (**4n**)

The sample was prepared from **3** (1.18 g, 5.2 mmol), imidazole (0.35 g, 5.2 mmol), and DBU (0.79 g, 5.2 mmol). The reaction time was 16 h. The obtained residue was purified by flash chromatography (eluent *n*-hexane/acetone, *v*/*v*, 4:1) to give **4n** (0.81 g, 53%) as a colorless oil. IR (ν_max_, cm^−1^): 1735 (C=O), 1697 (C=O). ^1^H-NMR (700 MHz, CDCl_3_): δ_H_ ppm 1.45 (s, 9H, C(CH_3_)_3_), 3.10 (s, 2H, CH_2_CO), 3.63 (s, 3H, COOCH_3_), 4.27–4.41 (m, 4H, Az 2,4-H), 6.94–6.99 (m, 1H, Imid CH), 7.08–6.10 (m, 1H, Imid CH), 7.60 (s, 1H, Imid CH). ^13^C-NMR (176 MHz, CDCl_3_): δ_C_ ppm 28.3 (C(CH_3_)_3_), 43.5 (CH_2_COOCH_3_), 52.2 (COOCH_3_), 53.4 (Az C-3), 60.1 (Az C-2,4), 80.7 (C(CH_3_)_3_), 116.9 (Imid CH), 130.1 (Imid CH), 135.6 (Imid CH), 155.9 (COOC(CH_3_)_3_), 169.2 (COOCH_3_). ^15^N-NMR (71 MHz, CDCl_3_): δ_N_ ppm –317.8 (Az), –199.2 (Imid N-1), –123.7 (Imid N-3). The HRMS (ESI^+^) for C_14_H_21_N_3_NaO4 ([M+Na]^+^) was calcd. as 318.1424 and found to be 318.1428.

##### *tert*-Butyl 3-(1*H*-benzimidazol-1-yl)-3-(2-methoxy-2-oxoethyl)azetidine-1-carboxylate (**4o**)

The sample was prepared from **3** (1.18 g, 5.2 mmol), benzimidazole (0.61 g, 5.2 mmol), and DBU (0.79 g, 5.2 mmol). The reaction time was 16 h. The obtained residue was purified by flash chromatography (eluent *n*-hexane/acetone, *v*/*v*, 4:1) to give **4o** (0.99 g, 56%) as a yellowish oil. IR (ν_max_, cm^−1^): 1725 (C=O), 1703 (C=O). ^1^H-NMR (700 MHz, CDCl_3_): δ_H_ ppm 1.45 (s, 9H, C(CH_3_)_3_), 3.26 (s, 2H, CH_2_CO), 3.55 (s, 3H, COOCH_3_), 4.45 (d, *J* = 9.2 Hz, 2H, Az 2,4-H_b_), 4.62 (d, *J* = 9.2 Hz, 2H, Az 2,4-H_a_), 7.11–7.15 (m, 1H, Bim CH), 7.26–7.34 (m, 2H, Bim 2 × CH), 7.81–7.87 (m, 1H, Bim CH), 8.05 (s, 1H, Bim CH). ^13^C-NMR (176 MHz, CDCl_3_): δ_C_ ppm 28.3 (C(CH_3_)_3_), 41.2 (CH_2_COOCH_3_), 52.0 (COOCH_3_), 53.1 (Az C-3), 59.0 (Az C-2,4), 80.7 (C(CH_3_)_3_), 109.9 (Bim CH), 121.2 (Bim CH), 122.6 (Bim CH), 123.3 (Bim CH), 131.8 (Bim Cq), 141.9 (Bim CH), 144.0 (Bim Cq), 155.9 (COOC(CH_3_)_3_), 169.3 (COOCH_3_). ^15^N-NMR (71 MHz, CDCl_3_): δ_N_ ppm –317.4 (Az), –220.9 (Bim N-1), –140.2 (Bim N-3). The HRMS (ESI^+^) for C_18_H_23_N_3_NaO_4_ ([M+Na]^+^) was calcd. as 358.1581 and found to be 368.1583.

##### *tert*-Butyl 3-(1*H*-indol-1-yl)-3-(2-methoxy-2-oxoethyl)azetidine-1-carboxylate (**4p**)

The sample was prepared from **3** (1.18 g, 5.2 mmol), indole (0.61 g, 5.2 mmol), and DBU (0.79 g, 5.2 mmol). The reaction time was 16 h. The obtained residue was purified by flash chromatography (eluent *n*-hexane/acetone, *v*/*v*, 4:1) to give **4p** (0.98 g, 55%) as a yellowish oil. IR (ν_max_, cm^−1^): 1735 (C=O), 1700 (C=O). ^1^H-NMR (700 MHz, CDCl_3_): δ_H_ ppm 1.45 (s, 9H, C(CH_3_)_3_), 3.20 (s, 2H, CH_2_CO), 3.50 (s, 3H, COOCH_3_), 4.44 (d, *J* = 8.9 Hz, 2H, Az 2,4-H_b_), 4.60 (d, *J* = 8.8 Hz, 2H, Az 2,4-H_a_), 6.50 (d, *J* = 3.2 Hz, 1H, Ind), 7.01 (d, *J* = 8.1 Hz, 1H, Ind), 7.13 (t, *J* = 7.4 Hz, 1H, Ind), 7.16–7.21 (m, 2H, Ind CH), 7.63 (d, *J* = 7.7 Hz, 1H, Ind CH). ^13^C-NMR (176 MHz, CDCl_3_): δ_C_ ppm 28.3 (C(CH_3_)_3_), 41.6 (CH_2_COOCH_3_), 51.8 (COOCH_3_), 53.9 (Az C-3), 58.8–59.2 (Az C-2,4), 80.4 (C(CH_3_)_3_), 102.1 (Ind CH), 109.9 (Ind CH), 120.0 (Ind CH), 121.7 (Ind CH), 121.9 (Ind CH), 126.1 (Ind CH), 129.4 (Ind Cq), 134.1 (Ind Cq), 156.2 (COOC(CH_3_)_3_), 169.8 (COOCH_3_). ^15^N-NMR (71 MHz, CDCl_3_): δ_N_ ppm –317.1 (Az), –238.0 (Ind N-1). The HRMS (ESI^+^) for C_19_H_24_N_2_NaO_4_ ([M+Na]^+^) was calcd. as 367.1628 and found to be 367.1633.

#### 3.2.3. General Procedure for Compounds **4q**–**s**

An appropriate *N*-heterocyclic compound (0.52 mmol), K_2_CO_3_ (72 mg, 0.52 mmol), and *tert*-butyl 3-(2-methoxy-2-oxoethylidene)azetidine-1-carboxylate **3** (118 mg, 0.52 mmol) were dissolved in acetonitrile (3.6 mL) and stirred at 65 °C for 16 h. The reaction was quenched by the addition of water (15 mL) and extracted with ethyl acetate (3 × 10 mL). The combined organic solutions were dried over anhydr. Na_2_SO_4_, and then the solvents were removed under reduced pressure. Purification was conducted via flash chromatography.

##### *tert*-Butyl 3-(2-methoxy-2-oxoethyl)-3-(1*H*-1,2,4-triazol-1-yl)azetidine-1-carboxylate (**4q**)

The sample was prepared from **3** (118 mg, 0.52 mmol), 1,2,4-triazole (61 mg, 0.52 mmol), and K_2_CO_3_ (72 mg, 0.52 mmol). The reaction time was 16 h. The obtained residue was purified by flash chromatography (eluent *n*-hexane/acetone, *v*/*v*, 4:1) to give **4q** (97 mg, 65%) as a yellowish oil. IR (ν_max_, cm^−1^): 1744 (C=O), 1700 (C=O). ^1^H-NMR (700 MHz, CDCl_3_): δ_H_ ppm 1.45 (s, 9H, C(CH_3_)_3_), 3.25 (s, 2H, CH_2_CO), 3.64 (s, 3H, COOCH_3_), 4.29 (d, *J* = 9.8 Hz, 2H, Az 2,4-H_b_), 4.43 (d, *J* = 9.7 Hz, 2H, Az 2,4-H_a_), 7.96 (s, 1H, Trz CH), 8.33 (s, 1H, Trz CH). ^13^C-NMR (176 MHz, CDCl_3_): δ_C_ ppm 28.3 (C(CH_3_)_3_), 41.6 (CH_2_COOCH_3_), 52.1 (COOCH_3_), 56.1 (Az C-3), 59.5 (Az C-2,4), 80.7 (C(CH_3_)_3_), 142.4 (Trz CH), 152.2 (Trz CH), 155.9 (COOC(CH_3_)_3_), 169.4 (COOCH_3_). ^15^N-NMR (71 MHz, CDCl_3_): δ_N_ ppm –317.3 (Az), –156.5 (Trz N-1), -132.7 (Trz N-4), –91.2 (Trz N-2). The HRMS (ESI^+^) for C_13_H_20_N_4_NaO_4_ ([M+Na]^+^) was calcd. as 319.1377 and found to be 319.1373.

##### *tert*-Butyl 3-(1*H*-benzotriazol-1-yl)-3-(2-methoxy-2-oxoethyl)azetidine-1-carboxylate (**4r**)

The sample was prepared from **3** (118 mg, 0.52 mmol), benzotriazole (70 mg, 0.52 mmol), and K_2_CO_3_ (72 mg, 0.52 mmol). The reaction time was 16 h. The obtained residue was purified by flash chromatography (eluent *n*-hexane/acetone, *v*/*v*, 4:1) to give **4r** (77 mg, 43%) as a colorless oil. IR (ν_max_, cm^−1^): 1732 (C=O), 1707 (C=O). ^1^H-NMR (700 MHz, CDCl_3_): δ_H_ ppm 1.48 (s, 9H, C(CH_3_)_3_), 3.43 (s, 2H, CH_2_CO), 3.58 (s, 3H, COOCH_3_), 4.55 (d, *J* = 9.8 Hz, 2H, Az 2,4-H_b_), 4.78 (d, *J* = 9.8 Hz, 2H, Az 2,4-H_a_), 7.41 (t, *J* = 7.7 Hz, 1H, Btz CH), 7.53 (t, *J* = 7.6 Hz, 1H, Btz CH), 7.64 (d, *J* = 8.4 Hz, 1H, Btz CH), 8.09 (d, *J* = 8.4 Hz, 1H, CH). ^13^C-NMR (176 MHz, CDCl_3_): δ_C_ ppm 28.3 (C(CH_3_)_3_), 42.2 (CH_2_COOCH_3_), 52.1 (COOCH_3_), 56.8 (Az C-3), 59.4 (Az C-2,4), 80.7 (C(CH_3_)_3_), 110.3 (Btz CH), 120.6 (Btz CH), 124.3 (Btz CH), 127.9 (Btz CH), 131.6 (Btz Cq), 146.6 (Btz Cq), 156.0 (COOC(CH_3_)_3_), 169.3 (COOCH_3_). ^15^N-NMR (71 MHz, CDCl_3_): δ_N_ ppm –316.9 (Az), –148.6 (Btz N-1), –42.0 (Btz N-3), –7.0 (Btz N-2). The HRMS (ESI^+^) for C_13_H_22_N_4_NaO_4_ ([M+Na]^+^) was calcd. as 369.1533 and found to be 369.1536.

##### *tert*-Butyl 3-(2*H*-benzotriazol-2-yl)-3-(2-methoxy-2-oxoethyl)azetidine-1-carboxylate (**4s**)

The sample was prepared from **3** (118 mg, 0.52 mmol), benzotriazole (70 mg, 0.52 mmol), and DBU (79 mg, 0.52 mmol). The reaction time was 16 h. The obtained residue was purified by flash chromatography (eluent *n*-hexane/acetone, *v*/*v*, 4:1) to give **4s** (59 mg, 33%) as a yellowish oil. IR (ν_max_, cm^−1^): 1740 (C=O), 1701 (C=O). ^1^H-NMR (700 MHz, CDCl_3_): δ_H_ ppm 1.48 (s, 9H, C(CH_3_)_3_), 3.48–3.61 (m, 2H, CH_2_CO), 3.63 (s, 3H, COOCH_3_), 4.39 (d, *J* = 9.6 Hz, 2H, Az 2,4-H_b_), 4.68–4.88 (m, 2H, Az 2,4-H_a_), 7.40 (dd, *J* = 6.7, 3.1 Hz, 2H, Btz 2 × CH), 7.87 (dd, *J* = 6.6, 3.1 Hz, 2H, Btz 2 × CH). ^13^C-NMR (176 MHz, CDCl_3_): δ_C_ ppm 28.3 (C(CH_3_)_3_), 41.2 (CH_2_COOCH_3_), 52.1 (COOCH_3_), 60.6 (Az C-2,4), 61.4 (Az C-3), 80.3 (C(CH_3_)_3_), 118.3 (Btz 2 × CH), 126.8 (Btz 2 × CH), 144.4 (Btz 2 × Cq), 156.2 (COOC(CH_3_)_3_), 169.4 (COOCH_3_). ^15^N-NMR (71 MHz, CDCl_3_): δ_N_ ppm –107.0 (Btz N-2), –69.1 (Btz N-1,3). The HRMS (ESI^+^) for C_15_H_20_F_3_N_3_NaO_4_ ([M+Na]^+^) was calcd. as 369.1533 and found to be 369.1530.

#### 3.2.4. General procedure for compounds **5a**–**n**

*tert*-Butyl 3-(4-bromo-1*H*-pyrazol-1-yl)-3-(2-methoxy-2-oxoethyl)azetidine-1-carboxylate **4k** (1.98 g, 5.3 mmol) was dissolved in dry dioxane (10 mL) and a current of argon was bubbled through the solution. After 20 min K_3_PO_4_ (3.38 g, 15.9 mmol), appropriate boronic acid (6.4 mmol) and tetrakis(triphenylphosphine)palladium (0.31 g, 0.27 mmol) were added. The reaction was stirred at 100 °C for 18 h. The reaction solution was quenched with water (10 mL) and extracted with ethyl acetate (3 × 15 mL). The organic layer was dried with anhydrous sodium sulphate, filtered, and concentrated under reduced pressure. The crude product was purified by flash chromatography using an eluent in the appropriate ratio.

##### *tert*-Butyl 3-(2-methoxy-2-oxoethyl)-3-(4-phenyl-1*H*-pyrazol-1-yl)azetidine-1-carboxylate (**5a**)

The sample was prepared from **4k** (1.98 g, 5.3 mmol), phenylboronic acid (0.78 g, 6.4 mmol), K_3_PO_4_ (3.38 g, 15.9 mmol), and tetrakis(triphenylphosphine)palladium (0.31 g, 0.27 mmol). The obtained residue was purified by flash chromatography (eluent *n*-hexane/acetone, *v*/*v*, 4:1) to give **5a** (1.85 g, 94%) as a yellowish oil. IR (ν_max_, cm^−1^): 1734 (C=O), 1688 (C=O). ^1^H-NMR (700 MHz, CDCl_3_): δ_H_ ppm 1.39 (s, 9H, C(CH_3_)_3_), 3.19 (s, 2H, CH_2_CO), 3.55 (s, 3H, OCH_3_), 4.24 (d, *J* = 9.6 Hz, 2H, Az 2,4-H_b_), 4.40 (d, *J* = 9.5 Hz, 2H, Az 2,4-H_a_), 7.17 (t, *J* = 7.4 Hz, 1H, Ph CH), 7.29 (t, *J* = 7.7 Hz, 2H, Ph 2 × CH), 7.41 (d, *J* = 7.6 Hz, 2H, Ph 2 × CH), 7.74 (s, 1H, Prz CH), 7.81 (s, 1H, Prz CH). ^13^C-NMR (176 MHz, CDCl_3_): δ_C_ ppm 28.3 (C(CH_3_)_3_), 42.1 (CH_2_COOCH_3_), 52.0 (COOCH_3_), 57.1 (Az C-3), 59.9 (Az C-2,4), 80.4 (C(CH_3_)_3_), 123.5, 124.7, 125.6, 126.7, 128.9, 132.2, 137.5, 156.1 (COOC(CH_3_)_3_), 169.9 (COOCH_3_). The HRMS (ESI^+^) for C_20_H_25_N_3_NaO_4_ ([M+Na]^+^) was calcd. as 394.1737 and found to be 394.1737.

##### *tert*-Butyl 3-(2-methoxy-2-oxoethyl)-3-[4-(4-methylphenyl)-1*H*-pyrazol-1-yl]azetidine-1-carboxylate (**5b**)

The sample was prepared from **4k** (1.98 g, 5.3 mmol), 4-methylphenylboronic acid (0.87 g, 6.4 mmol), K_3_PO_4_ (3.38 g, 15.9 mmol), and tetrakis(triphenylphosphine)palladium (0.31 g, 0.27 mmol). The obtained residue was purified by flash chromatography (eluent *n*-hexane/acetone, *v*/*v*, 4:1) to give **5b** (1.59 g, 78%) as a yellowish oil. IR (ν_max_, cm^−1^): 1733 (C=O), 1691 (C=O). ^1^H-NMR (700 MHz, CDCl_3_): δ_H_ ppm 1.38 (s, 9H, C(CH_3_)_3_), 2.28 (s, 3H, Ph-CH_3_), 3.17 (s, 2H, CH_2_CO), 3.54 (s, 3H, OCH_3_), 4.23 (d, *J* = 9.6 Hz, 2H, Az 2,4-H_b_), 4.39 (d, *J* = 9.6 Hz, 2H, Az 2,4-H_a_), 7.10 (d, *J* = 7.9 Hz, 2H, Ph 2 × CH), 7.30 (d, *J* = 8.1 Hz, 2H, Ph 2 × CH), 7.70 (s, 1H, Prz CH), 7.77 (s, 1H, Prz CH). ^13^C-NMR (176 MHz, CDCl_3_): δ_C_ ppm 21.1 (PhCH_3_), 28.3 (C(CH_3_)_3_), 42.1 (CH_2_COOCH_3_), 52.0 (COOCH_3_), 57.1 (Az C-3), 59.9 (Az C-2,4), 80.3 (C(CH_3_)_3_), 123.5, 124.4, 125.5, 129.3, 129.5, 136.3, 137.4, 156.1 (COOC(CH_3_)_3_), 169.9 (COOCH_3_). The HRMS (ESI^+^) for C_21_H_27_N_3_NaO_4_ ([M+Na]^+^) was calcd. as 408.1894 and found to be 408.1894.

##### *tert*-Butyl 3-(2-methoxy-2-oxoethyl)-3-[4-(2-methylphenyl)-1*H*-pyrazol-1-yl]azetidine-1-carboxylate (**5c**)

The sample was prepared from **4k** (1.98 g, 5.3 mmol), 2-methylphenylboronic acid (0.87 g, 6.4 mmol), K_3_PO_4_ (3.38 g, 15.9 mmol), and tetrakis(triphenylphosphine)palladium (0.31 g, 0.27 mmol). The obtained residue was purified by flash chromatography (eluent *n*-hexane/acetone, *v*/*v*, 4:1) to give **5c** (1.43 g, 70%) as a yellowish oil. IR (ν_max_, cm^−1^): 1736 (C=O), 1699 (C=O). ^1^H-NMR (700 MHz, CDCl_3_): δ_H_ ppm 1.39 (s, 9H, C(CH_3_)_3_), 2.31 (s, 3H, Ph-CH_3_), 3.18 (s, 2H, CH_2_CO), 3.56 (s, 3H, OCH_3_), 4.25 (d, *J* = 9.6 Hz, 2H, Az 2,4-H_b_), 4.42 (d, *J* = 9.6 Hz, 2H, Az 2,4-H_a_), 7.10–7.15 (m, 2H, Ph 2 × CH), 7.16–7.18 (m, 1H, Ph CH), 7.22–7.26 (m, 1H, Ph CH), 7.59 (s, 1H, Prz CH), 7.64 (s, 1H, Prz CH). ^13^C-NMR (176 MHz, CDCl_3_): δ_C_ ppm 21.2 (PhCH_3_), 28.3 (C(CH_3_)_3_), 42.2 (CH_2_COOCH_3_), 52.0 (COOCH_3_), 57.0 (Az C-3), 60.0 (Az C-2,4), 80.3 (C(CH_3_)_3_), 122.5, 126.0, 126.6, 127.0, 129.1, 130.7, 131.8, 135.3, 139.6, 156.1 (COOC(CH_3_)_3_), 169.9 (COOCH_3_). The HRMS (ESI^+^) for C_21_H_27_N_3_NaO_4_ ([M+Na]^+^) was calcd. as 408.1894 and found to be 408.1894.

##### *tert*-Butyl 3-(2-methoxy-2-oxoethyl)-3-[4-(4-methoxyphenyl)-1*H*-pyrazol-1-yl]azetidine-1-carboxylate (**5d**)

The sample was prepared from **4k** (1.98 g, 5.3 mmol), 4-methoxyphenylboronic acid (0.97 g, 6.4 mmol), K_3_PO_4_ (3.38 g, 15.9 mmol), and tetrakis(triphenylphosphine)palladium (0.31 g, 0.27 mmol). The obtained residue was purified by flash chromatography (eluent *n*-hexane/acetone, *v*/*v*, 4:1) to give **5d** (1.70 g, 80%) as a yellowish oil. IR (ν_max_, cm^−1^): 1735 (C=O), 1689 (C=O). ^1^H-NMR (700 MHz, CDCl_3_): δ_H_ ppm 1.38 (s, 9H, C(CH_3_)_3_), 3.18 (s, 2H, CH_2_CO), 3.55 (s, 3H, OCH_3_), 3.75 (s, 3H, Ph-OCH_3_), 4.23 (d, *J* = 9.6 Hz, 2H, Az 2,4-H_b_), 4.39 (d, *J* = 9.6 Hz, 2H, Az 2,4-H_a_), 6.83 (d, *J* = 8.8 Hz, 2H, Ph 2 × CH), 7.32 (d, *J* = 8.8 Hz, 2H, Ph 2 × CH), 7.66 (s, 1H, Pyr CH), 7.73 (s, 1H, Pyr CH). ^13^C-NMR (176 MHz, CDCl_3_): δ_C_ ppm 28.3 (C(CH_3_)_3_), 42.1 (CH_2_COOCH_3_), 52.0 (COOCH_3_), 55.3 (Ph-OCH_3_), 57.0 (Az C-3), 59.8 (Az C-2,4), 80.3 (C(CH_3_)_3_), 114.3, 123.3, 124.1, 124.8, 126.8, 137.3, 156.1 (COOC(CH_3_)_3_), 158.5 (Ph COCH_3_), 169.9 (COOCH_3_). The HRMS (ESI^+^) for C_21_H_27_N_3_NaO_5_ ([M+H]^+^) was calcd. as 402.2023 and found to be 402.2023.

##### *tert*-Butyl 3-(2-methoxy-2-oxoethyl)-3-[4-(3-methoxyphenyl)-1*H*-pyrazol-1-yl]azetidine-1-carboxylate (**5e**)

The sample was prepared from **4k** (1.98 g, 5.3 mmol), 3-methoxyphenylboronic acid (0.97 g, 6.4 mmol), K_3_PO_4_ (3.38 g, 15.9 mmol), and tetrakis(triphenylphosphine)palladium (0.31 g, 0.27 mmol). The obtained residue was purified by flash chromatography (eluent *n*-hexane/acetone, *v*/*v*, 4:1) to give **5e** (1.70 g, 80%) as a yellowish oil. IR (ν_max_, cm^−1^): 1736 (C=O), 1697 (C=O). ^1^H-NMR (700 MHz, CDCl_3_): δ_H_ ppm 1.39 (s, 9H, C(CH_3_)_3_), 3.19 (s, 2H, CH_2_CO), 3.55 (s, 3H, OCH_3_), 3.77 (s, 3H, Ph-OCH_3_), 4.24 (d, *J* = 9.6 Hz, 2H, Az 2,4-H_b_), 4.40 (d, *J* = 9.6 Hz, 2H, Az 2,4-H_a_), 6.72 (ddd, *J* = 8.3, 2.6, 0.9 Hz, 1H, Ph CH), 6.94 (dd, *J* = 2.6, 1.6 Hz, 1H, Ph CH), 7.00 (ddd, *J* = 7.6, 1.6, 1.0 Hz, 1H, Ph CH), 7.21 (t, *J* = 7.9 Hz, 1H, Ph CH), 7.72 (s, 1H, Prz CH), 7.80 (s, 1H, Prz CH). ^13^C-NMR (176 MHz, CDCl_3_): δ_C_ ppm 28.3 (C(CH_3_)_3_), 42.0 (CH_2_COOCH_3_), 52.0 (COOCH_3_), 55.3 (Ph-OCH_3_), 57.1 (Az C-3), 59.9 (Az C-2,4), 80.4 (C(CH_3_)_3_), 111.4, 112.0, 118.2, 123.4, 124.8, 129.9, 133.6, 137.6, 156.1 (COOC(CH_3_)_3_), 160.0 (Ph COCH_3_), 169.9 (COOCH_3_). The HRMS (ESI^+^) for C_21_H_27_N_3_NaO_5_ ([M+Na]^+^) was calcd. as 424.1843 and found to be 424.1843.

##### *tert*-Butyl 3-(2-methoxy-2-oxoethyl)-3-[4-(2-methoxyphenyl)-1*H*-pyrazol-1-yl]azetidine-1-carboxylate (**5f**)

The sample was prepared from **4k** (1.98 g, 5.3 mmol), 2-methoxyphenylboronic acid (0.97 g, 6.4 mmol), K_3_PO_4_ (3.38 g, 15.9 mmol), and tetrakis(triphenylphosphine)palladium (0.31 g, 0.27 mmol). The obtained residue was purified by flash chromatography (eluent *n*-hexane/acetone, *v*/*v*, 4:1) to give **5f** (0.62 g, 29%) as a yellowish oil. IR (ν_max_, cm^−1^): 1737 (C=O), 1698 (C=O). ^1^H-NMR (700 MHz, CDCl_3_): δ_H_ ppm 1.39 (s, 9H, C(CH_3_)_3_), 3.18 (s, 2H, CH_2_CO), 3.55 (s, 3H, OCH_3_), 3.84 (s, 3H, Ph-OCH_3_), 4.25 (d, *J* = 9.5 Hz, 2H, Az 2,4-H_b_), 4.43 (d, *J* = 9.5 Hz, 2H, Az 2,4-H_a_), 6.89 (d, *J* = 8.2 Hz, 1H, Ph × CH), 6.91 (d, *J* = 7.5 Hz, 1H, Ph × CH), 7.13–7.17 (m, 1H, Ph CH), 7.44 (dd, *J* = 7.6, 1.7 Hz, 1H, Ph CH), 7.86 (s, 1H, Prz CH), 7.99 (s, 1H, Prz CH). ^13^C-NMR (176 MHz, CDCl_3_): δ_C_ ppm 28.3 (C(CH_3_)_3_), 42.2 (CH_2_COOCH_3_), 51.9 (COOCH_3_), 55.4 (Ph-OCH_3_), 57.0 (Az C-3), 59.8 (Az C-2,4), 80.3 (C(CH_3_)_3_), 111.2, 118.9, 120.8, 121.0, 127.0, 127.5, 139.0, 155.8 (Ph COCH_3_), 156.2 (COOC(CH_3_)_3_), 169.9 (COOCH_3_). The HRMS (ESI^+^) for C_21_H_27_N_3_NaO_5_ ([M+Na]^+^) was calcd. as 424.1843 and found to be 424.1843.

##### *tert*-Butyl 3-[4-(4-fluorophenyl)-1*H*-pyrazol-1-yl]-3-(2-methoxy-2-oxoethyl)azetidine-1-carboxylate (**5g**)

The sample was prepared from **4k** (1.98 g, 5.3 mmol), 4-fluorophenylboronic acid (0.89 g, 6.4 mmol), K_3_PO_4_ (3.38 g, 15.9 mmol), and tetrakis(triphenylphosphine)palladium (0.31 g, 0.27 mmol). The obtained residue was purified by flash chromatography (eluent *n*-hexane/acetone, *v*/*v*, 4:1) to give **5g** (1.30 g, 63%) as a colorless oil. IR (ν_max_, cm^−1^): 1736 (C=O), 1697 (C=O). ^1^H-NMR (700 MHz, CDCl_3_): δ_H_ ppm 1.46 (s, 9H, C(CH_3_)_3_), 3.26 (s, 2H, CH_2_CO), 3.63 (s, 3H, OCH_3_), 4.31 (d, *J* = 9.6 Hz, 2H, Az 2,4-H_b_), 4.46 (d, *J* = 9.6 Hz, 2H, Az 2,4-H_a_), 7.03–7.09 (m, 2H, Ph 2 × CH), 7.44 (dd, *J* = 8.4, 5.4 Hz, 2H, Ph 2 × CH), 7.75 (s, 1H, Prz CH), 7.84 (s, 1H, Prz CH). ^13^C-NMR (176 MHz, CDCl_3_): δ_C_ ppm 28.3 (C(CH_3_)_3_), 42.0 (CH_2_COOCH_3_), 52.0 (COOCH_3_), 57.1 (Az C-3), 59.9 (Az C-2,4), 80.4 (C(CH_3_)_3_), 115.8 (d, *J* = 21.5 Hz), 122.6, 124.6, 127.2 (d, *J* = 7.8 Hz), 128.4 (d, *J* = 3.3 Hz), 137.4, 156.1 (COOC(CH_3_)_3_), 161.7 (d, *J* = 245.7 Hz), 169.9 (COOCH_3_). ^19^F-NMR (376 MHz, CDCl_3_): δ_F_ ppm –115.9 (tt, *J* = 9.3, 5.3 Hz). The HRMS (ESI^+^) for C_20_H_24_FN_3_NaO_4_ ([M+Na]^+^) was calcd. as 412.1643 and found to be 412.1645.

##### *tert*-Butyl 3-[4-(4-chlorophenyl)-1*H*-pyrazol-1-yl]-3-(2-methoxy-2-oxoethyl)azetidine-1-carboxylate (**5h**)

The sample was prepared from **4k** (1.98 g, 5.3 mmol), 4-chlorophenylboronic acid (1.01 g, 6.4 mmol), K_3_PO_4_ (3.38 g, 15.9 mmol), and tetrakis(triphenylphosphine)palladium (0.31 g, 0.27 mmol). The obtained residue was purified by flash chromatography (eluent *n*-hexane/acetone, *v*/*v*, 4:1) to give **5h** (1.14 g, 53%) as a colorless oil. IR (ν_max_, cm^−1^): 1736 (C=O), 1698 (C=O). ^1^H-NMR (700 MHz, CDCl_3_): δ_H_ ppm 1.46 (s, 9H, C(CH_3_)_3_), 3.26 (s, 2H, CH_2_CO), 3.63 (s, 3H, OCH_3_), 4.31 (d, *J* = 9.6 Hz, 2H, Az 2,4-H_b_), 4.46 (d, *J* = 9.6 Hz, 2H, Az 2,4-H_a_), 7.33 (d, *J* = 8.2 Hz, 2H, Ph 2 × CH), 7.41 (d, *J* = 8.4 Hz, 2H, Ph 2 × CH), 7.77 (s, 1H, Prz CH), 7.88 (s, 1H, Prz CH). ^13^C-NMR (176 MHz, CDCl_3_): δ_C_ ppm 28.3 (C(CH_3_)_3_), 42.0 (CH_2_COOCH_3_), 52.0 (COOCH_3_), 57.2 (Az C-3), 59.9 (Az C-2,4), 80.4 (C(CH_3_)_3_), 122.4, 124.8, 126.8, 129.0, 130.7 132.3, 137.4, 156.1 (COOC(CH_3_)_3_), 169.9 (COOCH_3_). The HRMS (ESI^+^) for C_20_H_24_ClN_3_NaO_4_ ([M+Na]^+^) was calcd. as 428.1348 and found to be 428.1351.

##### *tert*-Butyl 3-(2-methoxy-2-oxoethyl)-3-[4-(pyridin-4-yl)-1*H*-pyrazol-1-yl]azetidine-1-carboxylate (**5i**)

The sample was prepared from **4k** (200 mg, 0.53 mmol), 4-pyridinylboronic acid (79 mg, 0.64 mmol), K_3_PO_4_ (338 mg, 1.6 mmol), and tetrakis(triphenylphosphine)palladium (31 mg, 0.027 mmol). The obtained residue was purified by flash chromatography (eluent *n*-hexane/acetone, *v*/*v*, 4:1) to give **5i** (83 mg, 42%) as a colorless oil. IR (ν_max_, cm^−1^): 1735 (C=O), 1696 (C=O). ^1^H-NMR (700 MHz, CDCl_3_): δ_H_ ppm 1.46 (s, 9H, C(CH_3_)_3_), 3.28 (s, 2H, CH_2_CO), 3.64 (s, 3H, OCH_3_), 4.32 (d, *J* = 9.6 Hz, 2H, Az 2,4-H_b_), 4.47 (d, *J* = 9.7 Hz, 2H, Az 2,4-H_a_), 7.38 (d, *J* = 5.3 Hz, 2H, Pyr 2 × CH), 7.90 (s, 1H, Prz CH), 8.05 (s, Prz CH), 8.57 (d, *J* = 5.3 Hz, 2H, Pyr 2 × CH). ^13^C-NMR (176 MHz, CDCl_3_): δ_C_ ppm 28.3 (C(CH_3_)_3_), 41.9 (CH_2_COOCH_3_), 52.0 (COOCH_3_), 57.4 (Az C-3), 59.9 (Az C-2,4), 80.5 (C(CH_3_)_3_), 120.0, 120.8, 126.0, 137.8, 139.8, 150.3, 156.0 (COOC(CH_3_)_3_), 169.8 (COOCH_3_). ^15^N-NMR (71 MHz, CDCl_3_): δ_N_ ppm –316.9 (Az), –160.1 (Prz N-1), –77.6 (Prz N-2), –77.4 (Pyr). The HRMS (ESI^+^) for C_19_H_24_N_4_NaO_4_ ([M+Na]^+^) was calcd. as 395.1690 and found to be 395.1694.

##### *tert*-Butyl 3-(2-methoxy-2-oxoethyl)-3-[4-(pyridin-3-yl)-1*H*-pyrazol-1-yl]azetidine-1-carboxylate (**5j**)

The sample was prepared from **4k** (200 mg, 0.53 mmol), 4-pyridinylboronic acid (79 mg, 0.64 mmol), K_3_PO_4_ (338 mg, 1.6 mmol), and tetrakis(triphenylphosphine)palladium (31 mg, 0.027 mmol). The obtained residue was purified by flash chromatography (eluent *n*-hexane/acetone, *v*/*v*, 4:1) to give **5j** (79 mg, 40%) as a colorless oil. IR (ν_max_, cm^−1^): 1736 (C=O), 1697 (C=O). ^1^H-NMR (700 MHz, CDCl_3_): δ_H_ ppm 1.46 (s, 9H, C(CH_3_)_3_), 3.28 (s, 2H, CH_2_CO), 3.64 (s, 3H, OCH_3_), 4.32 (d, *J* = 9.7 Hz, 2H, Az 2,4-H_b_), 4.48 (d, *J* = 9.7 Hz, 2H, Az 2,4-H_a_), 7.30 (dd, *J* = 7.8, 4.8 Hz, 1H, Pyr CH), 7.74–7.78 (m, 1H, Pyr CH), 7.84 (s, 1H, Prz CH), 7.96 (s, 1H, Prz CH), 8.48 (d, *J* = 4.7 Hz, 1H, Pyr CH), 8.77 (d, *J* = 1.6 Hz, 1H, Pyr CH). ^13^C-NMR (176 MHz, CDCl_3_): δ_C_ ppm 28.3 (C(CH_3_)_3_), 42.0 (CH_2_COOCH_3_), 52.0 (COOCH_3_), 57.3 (Az C-3), 59.9 (Az C-2,4), 80.5 (C(CH_3_)_3_), 120.0, 123.7, 125.1, 128.2, 132.8, 137.5, 147.0, 147.8, 156.1 (COOC(CH_3_)_3_), 169.8 (COOCH_3_). ^15^N-NMR (71 MHz, CDCl_3_): δ_N_ ppm –316.9 (Az), –161.1 (Prz N-1), –78.2 (Prz N-2), –69.5 (Pyr). The HRMS (ESI^+^) for C_19_H_24_N_4_NaO_4_ ([M+Na]^+^) was calcd. as 395.1690 and found to be 395.1690.

##### *tert*-Butyl 3-(2-methoxy-2-oxoethyl)-3-[4-(6-methoxypyridin-3-yl)-1*H*-pyrazol-1-yl]azetidine-1-carboxylate (**5k**)

The sample was prepared from **4k** (1.98 g, 5.3 mmol), 6-methoxy-3-pyridinylboronic acid (0.98 g, 6.4 mmol), K_3_PO_4_ (3.38 g, 15.9 mmol), and tetrakis(triphenylphosphine)palladium (0.31 g, 0.27 mmol). The obtained residue was purified by flash chromatography (eluent *n*-hexane/acetone, *v*/*v*, 4:1) to give **5k** (0.79 g, 37%) as a white solid, mp 100.9–102.2 °C. IR (ν_max_, cm^−1^): 1738 (C=O), 1693 (C=O). ^1^H-NMR (700 MHz, CDCl_3_): δ_H_ ppm 1.46 (s, 9H, C(CH_3_)_3_), 3.27 (s, 2H, CH_2_CO), 3.63 (s, 3H, OCH_3_), 3.95 (s, 3H, Pyr OCH_3_), 4.32 (d, *J* = 9.8 Hz, 2H, Az 2,4-H_b_), 4.47 (d, *J* = 9.6 Hz, 2H, Az 2,4-H_a_), 6.76 (d, *J* = 8.5 Hz, 1H, Pyr CH), 7.66 (dd, *J* = 8.5, 2.5 Hz, 1H, Pyr CH), 7.74 (s, 1H, Prz CH), 7.84 (s, 1H, Prz CH), 8.29 (d, *J* = 2.5 Hz, 1H, Pyr CH). ^13^C-NMR (176 MHz, CDCl_3_): δ_C_ ppm 28.3 (C(CH_3_)_3_), 42.0 (CH_2_COOCH_3_), 52.0 (COOCH_3_), 53.5 (Pyr-OCH_3_), 57.2 (Az C-3), 60.0 (Az C-2,4), 80.4 (C(CH_3_)_3_), 110.9, 120.1, 121.6, 124.3, 136.3, 137.2, 143.6, 156.1 (COOC(CH_3_)_3_), 163.0 (Pyr COCH_3_), 169.9 (COOCH_3_). ^15^N-NMR (71 MHz, CDCl_3_): δ_N_ ppm –316.7 (Az), –162.3 (Prz N-1), –116.0 (Pyr), –79.0 (Prz N-2). The HRMS (ESI^+^) for C_20_H_26_N_4_NaO_5_ ([M+Na]^+^) was calcd. as 425.1795 and found to be 425.1795.

##### *tert*-Butyl 3-(2-methoxy-2-oxoethyl)-3-[4-(thiophen-3-yl)-1*H*-pyrazol-1-yl]azetidine-1-carboxylate (**5l**)

The sample was prepared from **4k** (200 mg, 0.53 mmol), 3-thienylboronic acid (82 mg, 0.64 mmol), K_3_PO_4_ (338 mg, 1.6 mmol), and tetrakis(triphenylphosphine)palladium (31 mg, 0.027 mmol). The obtained residue was purified by flash chromatography (eluent *n*-hexane/acetone, *v*/*v*, 4:1) to give **5l** (128 mg, 64%) as a yellowish oil. IR (ν_max_, cm^−1^): 1735 (C=O), 1696 (C=O). ^1^H-NMR (700 MHz, CDCl_3_): δ_H_ ppm 1.46 (s, 9H, C(CH_3_)_3_), 3.25 (s, 2H, CH_2_CO), 3.63 (s, 3H, OCH_3_), 4.30 (d, *J* = 9.6 Hz, 2H, Az 2,4-H_b_), 4.46 (d, *J* = 9.6 Hz, 2H, Az 2,4-H_a_), 7.19–7.22 (m, 1H, Thio CH), 7.24–7.26 (m, 1H, Thio CH), 7.33–7.36 (m, 1H, Thio CH) 7.73 (s, 1H, Prz CH), 7.81 (s, 1H, Prz CH). ^13^C-NMR (176 MHz, CDCl_3_): δ_C_ ppm 28.3 (C(CH_3_)_3_), 42.0 (CH_2_COOCH_3_), 52.0 (COOCH_3_), 57.1 (Az C-3), 59.9 (Az C-2,4), 80.4 (C(CH_3_)_3_), 118.6, 118.9, 124.7, 126.1, 126.2, 132.9, 137.8, 156.1 (COOC(CH_3_)_3_), 169.9 (COOCH_3_). The HRMS (ESI^+^) for C_18_H_23_N_3_NaO_4_S ([M+Na]^+^) was calcd. as 400.1301 and found to be 400.1304.

##### *tert*-Butyl 3-(2-methoxy-2-oxoethyl)-3-[4-(4-methylthiophen-3-yl)-1*H*-pyrazol-1-yl]azetidine-1-carboxylate (**5m**)

The sample was prepared from **4k** (200 mg, 0.53 mmol), (4-methylthiophen-3-yl)boronic acid (91 mg, 0.64 mmol), K_3_PO_4_ (338 mg, 1.6 mmol), and tetrakis(triphenylphosphine)palladium (31 mg, 0.027 mmol). The obtained residue was purified by flash chromatography (eluent *n*-hexane/acetone, *v*/*v*, 4:1) to give **5m** (129 mg, 62%) as a yellowish oil. IR (ν_max_, cm^−1^): 1735 (C=O), 1700 (C=O). ^1^H-NMR (700 MHz, CDCl_3_): δ_H_ ppm 1.46 (s, 9H, C(CH_3_)_3_), 2.31 (s, 3H, CH_3_), 3.25 (s, 2H, CH_2_CO), 3.63 (s, 3H, OCH_3_), 4.31 (d, *J* = 9.6 Hz, 2H, Az 2,4-H_b_), 4.48 (d, *J* = 9.6 Hz, 2H, Az 2,4-H_a_), 6.97–7.00 (m, 1H, Thio CH), 7.19 (d, *J* = 3.3 Hz, 1H, Thio CH), 7.68 (s, 1H, Prz CH), 7.73 (s, 1H, Prz CH). ^13^C-NMR (176 MHz, CDCl_3_): δ_C_ ppm 15.9 (Thio-CH_3_), 28.3 (C(CH_3_)_3_), 42.1 (CH_2_COOCH_3_), 51.9 (COOCH_3_), 57.0 (Az C-3), 59.8 (Az C-2,4), 80.3 (C(CH_3_)_3_), 118.2, 121.2, 122.1, 125.4, 133.1, 135.9, 138.8, 156.1 (COOC(CH_3_)_3_), 169.9 (COOCH_3_). The HRMS (ESI^+^) for C_19_H_25_N_3_NaO_4_S ([M+Na]^+^) was calcd. as 414.1458 and found to be 414.1462.

##### *tert*-Butyl 3-(4-cyclopropyl-1H-pyrazol-1-yl)-3-(2-methoxy-2-oxoethyl)azetidine-1-carboxylate (**5n**)

Method I. To a solution of *tert*-butyl 3-(4-bromo-1*H*-pyrazol-1-yl)-3-(2-methoxy-2-oxoethyl)azetidine-1-carboxylate **4k** (100 mg, 0.26 mmol) in toluene (5 mL) under an argon atmosphere, cyclopropylboronic acid (30 mg, 0.34 mmol), K_3_PO_4_ (193 mg, 0.9 mmol), and tetrakis(triphenylphosphine)palladium (30 mg, 0.026 mmol) were added, and the reaction mixture was refluxed for 18 h. After completion of the reaction as monitored by TLC, the mixture was quenched with water (10 mL) and extracted with ethyl acetate (3 × 10 mL). The organic layers were combined, washed with brine, and dried over Na_2_SO_4_, filtrated, and the solvent was evaporated. The obtained residue was purified by flash chromatography (eluent *n*-hexane/acetone, *v*/*v*, 4:1) to give **5n** (27 mg, 31%) as a clear oil. IR (ν_max_, cm^−1^): 1738 (C=O), 1699 (C=O). ^1^H-NMR (700 MHz, CDCl_3_): δ_H_ ppm 0.41–0.45 (m, 2H, Cpr CH_2_), 0.74–0.78 (m, 2H, Cpr CH_2_), 1.38 (s, 9H, C(CH_3_)_3_), 1.58–1.62 (m, 1H, Cpr CH), 3.11 (s, 2H, CH_2_CO), 3.55 (s, 3H, OCH_3_), 4.18 (d, *J* = 9.5 Hz, 2H, Az 2,4-H_b_), 4.31 (d, *J* = 9.5 Hz, 2H, Az 2,4-H_a_), 7.23 (s, 1H, Prz CH), 7.30 (s, 1H, Prz CH). ^13^C-NMR (176 MHz, CDCl_3_): δ_C_ ppm 5.3 (Cpr CH), 7.6 (Cpr 2 × CH_2_), 28.2 (C(CH_3_)_3_), 41.8 (CH_2_COOCH_3_), 51.6 (COOCH_3_), 56.6 (Az C-3), 59.6 (Az C-2,4), 79.8 (C(CH_3_)_3_), 124.8 (Prz Cq), 124.9 (Prz CH), 138.1 (Prz CH), 155.9 (COOC(CH_3_)_3_), 169.7 (COOCH_3_). The HRMS (ESI^+^) for C_17_H_25_N_3_NaO_4_ ([M+Na]^+^) was calcd. as 358.1737 and found to be 358.1737.

Method II. To a solution of *tert*-butyl 3-(4-bromo-1*H*-pyrazol-1-yl)-3-(2-methoxy-2-oxoethyl)azetidine-1-carboxylate **4k** (100 mg, 0.26 mmol) in toluene (5 mL) under an argon atmosphere, cyclopropylboronic acid (30 mg, 0.34 mmol), K_3_PO_4_ (193mg, 0.9 mmol), P(cHex)_3_ (10 mg, 0.026 mmol), and Pd(OAc)_2_ (6 mg, 0.026 mmol) were added, and the reaction mixture was refluxed for 18 h. After completion of the reaction as monitored by TLC, the mixture was quenched with water (10 mL) and extracted with ethyl acetate (3 × 10 mL). The organic layers were combined, washed with brine, dried over Na_2_SO_4_, and filtrated, and the solvent was evaporated. The obtained residue was purified by flash chromatography (eluent *n*-hexane/acetone, *v*/*v*, 4:1) to give **5n** (56 mg, 64%) as a clear oil.

#### 3.2.5. Methyl(oxetan-3-ylidene)acetate (**7**)

Neat trimethyl phosphonoacetate **2** (3.79 g, 20.8 mmol) was added to a cooled (0°C) suspension of NaH (60% dispersion in mineral oil) (0.83 g, 20.8 mmol) in dry THF (70 mL). After 20 min, a solution of 3-oxetanone **6** (1.50 g, 20.8 mmol) in dry THF (20 mL) was added, and the resulting mixture was stirred for 1 h. The reaction was quenched by the addition of water (70 mL). The organic layer was separated, and the aqueous one was extracted with ethyl acetate (3 × 70 mL). The combined organic solutions were dried over anhydr. Na_2_SO_4_, and then the solvents were removed under reduced pressure. The obtained residue was purified by flash chromatography (eluent *n*-hexane/acetone, *v*/*v*, 4:1) to give **7** (1.95 g, 73%) as a white solid, mp 50.8–51.6 °C. IR (ν_max_, cm^−1^): 1716 (C=O), 1207 (C-O-C). ^1^H-NMR (700 MHz, CDCl_3_): δ_H_ ppm 3.72 (s, 3H, OCH_3_), 5.29–5.32 (m, 2H, Ox 2,4-H), 5.49–5.53 (m, 2H, Ox 2,4-H), 5.55–5.67 (m, 1H, CHCO). ^13^C-NMR (176 MHz, CDCl_3_): δ_C_ ppm 51.5 (COOCH_3_), 78.5 (Ox C-2,4), 81.1 (C(CH_3_)_3_), 110.7 (Ox C-3), 159.6 (COOC(CH_3_)_3_), 165.7 (COOCH_3_). The HRMS (ESI^+^) for C_6_H_8_NaO_3_ ([M+Na]^+^) was calcd. as 151.0366 and found to be 151.0366.

#### 3.2.6. General Procedure for Compounds **8a**–**g**

An appropriate *N*-heterocyclic compound (11.7 mmol), DBU (1.78 g, 11.7 mmol), and methyl 2-(oxetan-3-ylidene)acetate **7** (1.50 g, 11.7 mmol) were dissolved in acetonitrile (4 mL) and stirred at 45 °C for 24 h. The reaction was quenched by the addition of water (15 mL) and extracted with ethyl acetate (3 × 10 mL). The combined organic solutions were dried over anhydr. Na_2_SO_4_, and then the solvents were removed under reduced pressure. Purification was conducted via flash chromatography.

##### Methyl(3-{3-[(*tert*-butoxycarbonyl)amino]azetidin-1-yl}oxetan-3-yl)acetate (**8a**)

The sample was prepared from **7** (0.33 g, 2.6 mmol), 3-*N*-Boc-aminoazetidine hydrochloride (0.54 g, 2.6 mmol), and DBU (0.4 g, 2.6 mmol). The obtained residue was purified by flash chromatography (eluent *n*-hexane/ethyl acetate, *v*/*v*, 2:1 and dichloromethane/methanol, v/v, 100:1) to give 8a (0.55 g, 71%) as a slightly yellow oil. IR (ν_max_, cm^−1^): 3314 (N-H), 1719 (C=O), 1162 (C-O-C). ^1^H-NMR (700 MHz, CDCl_3_): δ_H_ ppm 1.44 (s, 9H, C(CH_3_)_3_), 2.66 (s, 2H, CH_2_CO), 3.11–3.24 (m, 2H, Az CH_2_), 3.64–3.71 (m, 5H, Az CH_2_, OCH_3_), 4.10–4.35 (m, 1H, Az CH), 4.57 (d, *J* = 7.2 Hz, 2H, Ox 2,4-H), 4.71 (d, *J* = 7.2 Hz, 2H, Ox 2,4-H), 4.98–5.06 (m, 1H, NH). ^13^C-NMR (176 MHz, CDCl_3_): δ_C_ ppm 28.4 (C(CH_3_)_3_), 40.1 (CH_2_COOCH_3_), 40.7 (Az C-3), 51.7 (COOCH_3_), 55.7 (Az C-2,4), 61.7 (Ox C-3), 76.0 (Ox C-2,4), 79.7 (C(CH_3_)_3_), 155.0 (COOC(CH_3_)_3_), 171.0 (COOCH_3_). ^15^N-NMR (71 MHz, CDCl_3_): δ_N_ ppm –348.4 (Az), –288.5 (NH). The HRMS (ESI^+^) for C_14_H_24_N_2_O_5_ ([M+H]^+^) was calcd. as 301.1764 and found to be 301.1758.

##### Methyl(3-{(3*S*)-3-[(*tert*-butoxycarbonyl)amino]pyrrolidin-1-yl}oxetan-3-yl)acetate (**8b**)

The sample was prepared from **7** (1.50 g, 11.7 mmol), (*S*)-3-Boc-aminopyrrolidine (2.18 g, 11.7 mmol), and DBU (1.78 g, 11.7 mmol). The obtained residue was purified by flash chromatography (eluent ethyl acetate) to give **8b** (2.39 g, 65%) as a slightly yellow oil. [α]_D_^20^ = –15.4 (*c* 0.87, MeOH). IR (ν_max_, cm^−1^) 3330 (N-H), 1735 (C=O), 1706 (C=O), 1165 (C-O-C). ^1^H-NMR (700 MHz, CDCl_3_): δ_H_ ppm 1.44 (s, 9H, C(CH_3_)_3_), 1.60–1.73 (m, 1H, Pyrr 4-H C*H_a_*H_b_), 2.14–2.27 (m, 1H, Pyrr 4-H CH_a_*H_b_*), 2.63–2.75 (m, 2H, Pyrr 2-H C*H_a_*H_b_ and 5-H C*H_a_*H_b_) 2.80–2.92 (m, 3H, Pyrr 2-H CH_a_*H_b_* and CH_2_CO), 2.93–3.00 (m, 1H, Pyrr 5-H CH_a_*H_b_*), 3.69 (s, 3H, OCH_3_), 3.97–4.24 (m, 1H, Pyrr 3-H), 4.55 (d, *J* = 6.9 Hz, 1H, Ox 2,4-H), 4.62 (d, *J* = 6.9 Hz, 1H, Ox 2,4-H), 4.76 (dd, *J* = 6.9, 4.4 Hz, 2H, Ox 2,4-H), 4.78–4.85 (m, 1H, NH). ^13^C-NMR (176 MHz, CDCl_3_): δ_C_ ppm 28.4 (C(CH_3_)_3_), 32.1 (Pyrr C-4), 40.6 (CH_2_COOCH_3_), 45.1 (Pyrr C-5), 49.8 (Pyrr C-3), 51.7 (COOCH_3_), 53.7 (Pyrr C-2), 61.2 (Ox C-3), 77.4 (Ox C-2,4), 77.8 (Ox C-2,4), 79.3 (C(CH_3_)_3_), 155.3 (COOC(CH_3_)_3_), 171.3 (COOCH_3_). ^15^N-NMR (71 MHz, CDCl_3_): δ_N_ ppm –330.6 (Pyrr), –284.3 (NH). The HRMS (ESI^+^) for C_15_H_26_N_2_O_5_ ([M+H]^+^) was calcd. as 315.1914 and found to be 315.1915.

##### Methyl(3-{(3*R*)-3-[(*tert*-butoxycarbonyl)amino]pyrrolidin-1-yl}oxetan-3-yl)acetate (**8c**)

The sample was prepared from **7** (0.30 g, 2.3 mmol), ®-3-Boc-aminopyrrolidine (0.43 g, 2.3 mmol), and DBU (0.36 g, 2.3 mmol). The obtained residue was purified by flash chromatography (eluent ethyl acetate) to give **8c** (1.28 g, 87%) as a slightly yellow oil. [α]_D_^20^ = 15.6 (*c* 0.87, MeOH). IR (ν_max_, cm^−1^) 3331 (N-H), 1736 (C=O), 1706 (C=O), 1165 (C-O-C). ^1^H-NMR (700 MHz, CDCl_3_): δ_H_ ppm 1.44 (s, 9H, C(CH_3_)_3_), 1.63–1.70 (m, 1H, Pyrr 4-H C*H_a_*H_b_), 2.14–2.24 (m, 1H, Pyrr 4-H CH_a_*H_b_*), 2.65–2.72 (m, 2H, Pyrr 2-H C*H_a_*H_b_ and 5-H C*H_a_*H_b_), 2.81–2.90 (m, 3H, Pyrr 2-H CH_a_*H_b_* and CH_2_CO), 2.93–3.02 (m, 1H, Pyrr 5-H CH_a_*H_b_*), 3.69 (s, 3H, OCH_3_), 4.00–4.22 (m, 1H, Pyrr 3-H), 4.55 (d, *J* = 6.9 Hz, 1H, Ox 2,4-H), 4.62 (d, *J* = 6.9 Hz, 1H, Ox 2,4-H), 4.75 (dd, *J* = 6.9, 4.5 Hz, 2H, Ox 2,4-H), 4.79–4.88 (m, 1H, NH). ^13^C-NMR (176 MHz, CDCl_3_): δ_C_ ppm 28.4 (C(CH_3_)_3_), 32.1 (Pyrr C-4), 40.6 (CH_2_COOCH_3_), 45.1 (Pyrr C-5), 49.8 (Pyrr C-3), 51.7 (COOCH_3_), 53.7 (Pyrr C-2), 61.2 (Ox C-3), 77.4 (Ox C-2,4), 77.8 (Ox C-2,4), 79.3 (C(CH_3_)_3_), 155.3 (COOC(CH_3_)_3_), 171.3 (COOCH_3_). The HRMS (ESI^+^) for C_15_H_26_N_2_O_5_ ([M+H]^+^) was calcd. as 315.1914 and found to be 315.1915.

##### Methyl(3-{(3*S*)-3-[(*tert*-butoxycarbonyl)amino]piperidin-1-yl}oxetan-3-yl)acetate (**8d**) 

The sample was prepared from **7** (1.50 g, 11.7 mmol), (*S*)-3-Boc-aminopiperidine (2.34 g, 11.7 mmol), and DBU (1.78 g, 11.7 mmol). The obtained residue was purified by flash chromatography (eluent ethyl acetate) to give **8d** (1.92 g, 50%) as a clear oil. [α]_D_^20^ = –21.8 (*c* 0.87, MeOH). IR (ν_max_, cm^−1^) 3331 (N-H), 1731 (C=O), 1706 (C=O), 1163 (C-O-C). ^1^H-NMR (700 MHz, CDCl_3_): δ_H_ ppm 1.45 (s, 9H, C(CH_3_)_3_), 1.51–1.76 (m, 4H, Pip 4-H and Pip 5-H), 2.21–2.36 (m, 2H, Pip 6-H C*H_a_*H_b_ and 2-H C*H_a_*H_b_), 2.37–2.57 (m, 2H, Pip 6-H CH_a_*H_b_* and 2-H CH_a_*H_b_*), 2.71 (m, 2H, CH_2_CO), 3.71 (s, 3H, OCH_3_), 3.72–3.79 (m, 1H, Pip 3-H), 4.50–4.64 (m, 4H, Ox 2,4-H), 5.07 (s, 1H, NH). ^13^C-NMR (176 MHz, CDCl_3_): δ_C_ ppm 22.4 (Pip C-5), 28.5 (C(CH_3_)_3_), 29.5 (Pip C-4), 34.7 (CH_2_COOCH_3_), 45.9 (Pip C-3), 46.2 (Pip C-6), 51.3 (Pip C-2), 51.9 (COOCH_3_), 62.2 (Ox C-3), 79.2 (Ox C-2,4), 79.3 (Ox C-2,4), 79.5 (C(CH_3_)_3_), 155.1 (COOC(CH_3_)_3_), 172.1 (COOCH_3_). ^15^N-NMR (71 MHz, CDCl_3_): δ_N_ ppm –331.1 (Pip), –291.2 (NH). The HRMS (ESI^+^) for C_16_H_28_N_2_O_5_ ([M+H]^+^) was calcd. as 329.2071 and found to be 329.2071.

##### Methyl(3-{(3*R*)-3-[(*tert*-butoxycarbonyl)amino]piperidin-1-yl}oxetan-3-yl)acetate (**8e**)

The sample was prepared from **7** (1.50 g, 11.7 mmol), (*R*)-3-Boc-aminopiperidine (2.34 g, 11.7 mmol), and DBU (1.78 g, 11.7 mmol). The obtained residue was purified by flash chromatography (eluent *n*-hexane/ethyl acetate, *v*/*v*, 1:1) to give **8e** (2.11 g, 55%) as a clear oil. [α]_D_^20^ = 22.1 (*c* 0.87, MeOH). IR (ν_max_, cm^−1^) 3330 (N-H), 1732 (C=O), 1706 (C=O), 1163 (C-O-C). ^1^H-NMR (700 MHz, CDCl_3_): δ_H_ ppm 1.45 (s, 9H, C(CH_3_)_3_), 1.51–1.74 (m, 4H, Pip 4-H and Pip 5-H), 2.19–2.35 (m, 2H, Pip 6-H C*H_a_*H_b_ and 2-H C*H_a_*H_b_), 2.37–2.57 (m, 2H, Pip 6-H CH_a_*H_b_* and 2-H CH_a_*H_b_*), 2.71 (m, 2H, CH_2_CO), 3.71 (s, 3H, OCH_3_), 3.73–3.78 (m, 1H, Pip 3-H), 4.46–4.63 (m, 4H, Ox 2,4-H), 5.08 (s, 1H, NH). ^13^C-NMR (176 MHz, CDCl_3_): δ_C_ ppm 22.4 (Pip C-5), 28.5 (C(CH_3_)_3_), 29.5 (Pip C-4), 34.7 (CH_2_COOCH_3_), 45.9 (Pip C-3), 46.2 (Pip C-6), 51.3 (Pip C-2), 51.9 (COOCH_3_), 62.2 (Ox C-3), 79.2 (Ox C-2,4), 79.3 (Ox C-2,4), 79.5 (C(CH_3_)_3_), 155.1 (COOC(CH_3_)_3_), 172.1 (COOCH_3_). ^15^N-NMR (71 MHz, CDCl_3_): δ_N_ ppm –331.1 (Pip), –291.0 (NH). The HRMS (ESI^+^) for C_16_H_28_N_2_O_5_ ([M+H]^+^) was calcd. as 329.2071 and found to be 329.2071.

##### Methyl(3-{4-[(*tert*-butoxycarbonyl)amino]piperidin-1-yl}oxetan-3-yl)acetate (**8f**)

The sample was prepared from **7** (0.75 g, 5.75 mmol), 4-Boc-aminopiperidine (1.17 g, 5.75 mmol), and DBU (0.89 g, 5.75 mmol). The obtained residue was purified by flash chromatography (eluent dichloromethane/methanol, *v*/*v*, 100:1) to give **8f** (1.10 g, 58%) as a clear oil. IR (ν_max_, cm^−1^) 3308 (N-H), 1725 (C=O), 1679 (C=O), 1164 (C-O-C). ^1^H-NMR (700 MHz, CDCl_3_): δ_H_ ppm 1.35–1.51 (m, 11H, C(CH_3_)_3_, Pip CH_2_), 1.82–1.97 (m, 2H, Pip CH_2_), 2.18 (td, *J* = 11.3, 2.5 Hz, 2H, Pip CH_2_), 2.60 (dt, *J* = 11.3, 3.8 Hz 2H, Pip CH_2_), 2.72 (s, 2H, CH_2_CO), 3.22–3.52 (m, 1H, Pip CH), 3.70 (s, 3H, OCH_3_), 4.55 (s, 4H, Ox 2 × CH_2_), 4.57–4.66 (m, 1H, NH). ^13^C-NMR (176 MHz, CDCl_3_): δ_C_ ppm 28.4 (C(CH_3_)_3_), 32.9 (Pip 2 × CH_2_), 34.3 (CH_2_COOCH_3_), 44.6 (Pip 2 × CH_2_), 47.8 (Pip CH), 51.9 (COOCH_3_), 62.3 (Ox C-3), 79.2 (Ox C-2,4), 79.4 (C(CH_3_)_3_), 155.1 (COOC(CH_3_)_3_), 172.2 (COOCH_3_). The HRMS (ESI^+^) for C_16_H_28_N_2_O_5_ ([M+Na]^+^) was calcd. as 351.1898 and found to be 351.1890.

##### Methyl[3-(4-{[(*tert*-butoxycarbonyl)amino]methyl}piperidin-1-yl)oxetan-3-yl]acetate (**8g**)

The sample was prepared from **7** (1.5 g, 11.7 mmol), 4-Boc-aminopiperidine (2.51 g, 11.7 mmol), and DBU (1.78 g, 11.7 mmol). The obtained residue was purified by flash chromatography (eluent dichloromethane/methanol, *v*/*v*, 100:3) to give **8g** (2.20 g, 55%) as a clear oil. IR (ν_max_, cm^−1^) 3374 (N-H), 1723 (C=O), 1681 (C=O), 1165 (C-O-C). ^1^H-NMR (700 MHz, CDCl_3_): δ_H_ ppm 1.23 (qd, *J* = 11.8, 3.8 Hz, 2H, Pip CH_2_), 1.44 (s, 9H, C(CH_3_)_3_), 1.63–1.74 (m, 2H, Pip CH_2_), 2.08 (td, *J* = 11.5, 2.4 Hz, 2H, Pip CH_2_), 2.62 (dt, *J* = 11.9, 3.5 Hz, 2H, Pip CH_2_), 2.72 (s, 2H, CH_2_CO), 3.01 (t, *J* = 6.4 Hz, 2H, CH_2_), 3.70 (s, 3H, OCH_3_), 4.52–4.62 (m, 4H, 2 × CH_2_, Ox), 4.60–4.71 (m, 1H, NH).^13^C-NMR (176 MHz, CDCl_3_): δ_C_ ppm 28.4 (C(CH_3_)_3_), 30.1 (Pip 2 × CH_2_), 34.2 (CH_2_COOCH_3_), 36.5 (Pip CH), 45.5 (Pip 2 × CH_2_), 46.1 (CH_2_NH), 51.9 (COOCH_3_), 62.4 (Ox C-3), 79.1 (C(CH_3_)_3_), 79.4 (Ox C-2,4), 156.0 (COOC(CH_3_)_3_), 172.4 (COOCH_3_). The HRMS (ESI^+^) for [C_17_H_30_N_2_O_5_ ([M+H]^+^) was calcd. as 343.2234 and found to be 343.2228.

## 4. Conclusions

We developed a general approach for the preparation of new heterocyclic amino-acid-like building blocks containing azetidine and oxetane rings through aza-Michael addition, starting from methyl 2-(azetidin-3-ylidene)- and methyl 2-(oxetan-3-ylidene)acetates with heterocyclic aliphatic and heterocyclic aromatic amines. The synthesis and diversification of the azetidine building blocks were achieved through palladium-catalysed Suzuki–Miyaura cross-coupling reactions from a corresponding brominated pyrazole scaffold with alkyl and aryl boronic acids. These new heterocyclic compounds could be reliably determined using advanced NMR spectroscopy techniques, in particular by conducting ^1^H-^1^H NOESY, ^1^H-^13^C HMBC, and ^1^H-^15^N HMBC experiments.

## Data Availability

The data presented in this study are available on request from the corresponding authors. ^1^H and ^13^C NMR data are available via Mendeley Data, V1, doi:10.17632/5jh3p4v6p5.1.

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
