# Peer review of "Synthesis of New Azetidine and Oxetane Amino Acid Derivatives through Aza-Michael Addition of NH-Heterocycles with Methyl 2-(Azetidin- or Oxetan-3-Ylidene)Acetates"

_molecules, 2023, doi:10.3390/molecules28031091_

Round 1

Reviewer 1 Report

The article entitled “Synthesis of new azetidine and oxetane amino acid derivatives through aza-Michael addition of NH-heterocycles with methyl 2-(azetidin- or oxetan-3-ylidene)acetates” by Gudelis E. et al, describes an alternative protocol for the synthesis of amino ester derivatives containing azetidine or oxetane in its structure. The developed protocol proved to be efficient for such synthesis. Satisfactorily, the desired products were obtained in moderate to excellent yields, showing the good efficiency of the method and proving the wide tolerance of the substrate. In summary, some major revisions are needed:

- Figure 1 molecular representations XI and XII the oxygen atom is making 3 bonds and contains no charge, in reference 30 of the article the representation of the molecule is not the same as figure 1.

- In the body of the text where there is a discussion of Figure 1, representations IX, X, XI, XII, XIII (page 2 and page 3), the terms “acetic acid” and “amino acid” are described and used, in in my view, it is more appropriate to update the terms “acetate” and “amino esters” because there is no OH group in these representations.

- The beginning of Results and Discussions the text is written in such a way that there are many reports of works by other research groups, for a more dynamic reading I suggest that you optimize these reports.

- In Table 2, the term “catalyst” is used for reagents in an amount of 1 equiv., a very large amount to say that it is a catalyst, I suggest changing this term to another more appropriate one.

-The description of the high masses of some compounds have the same calculated and found values, however the SI says the opposite, for example, compound 4e and the exact mass values calculated and found are the same, however in the SI the value found is 313.2122 . I suggest a complete overhaul.

Additional suggestions,

- For greater reliability of the NMR data, it is interesting to carry out peaks by peaks, at values of TMS = 0.0 ppm (for 1H NMR) and CDCl3 = 77.0 ppm (for 13C NMR).

- Examples containing hydroxyls, the H atom of this functional group does not appear in the 1H NMR, if possible, a zoom in this region would be interesting.

Reviewer 2 Report

The authors reports the synthetic approach to azetidine derivatives. The reaction is known, but it was perfectly applied to the synthesis of this very important compounds. The authors provided good introduction with appropriate references, the significant scope for all the transformations and detailed structural study using NMR-techniques. I also did not find any typos or drawbacks, so I consider this perfect manuscript can be published in present form.

Author Response

We would like to thank the Reviewer for kind evaluation of our Manuscript.

Reviewer 3 Report

The authors have been submitted an article in simple synthesis and characterization of new azetidine and oxetane amino acid derivatives from commercially available starting compound entitled

Synthesis of new azetidine and oxetane amino acid derivatives through aza-Michael addition of NH-heterocycles with methyl 2-(azetidin- or oxetan-3-ylidene)acetates

I recommend that this article be published in the journal Molecules on the condition that the following minor points are addressed. The topic of the thesis is within the scope of the journal, with particular reference to the synthesis of small heterocyclic compounds, which may be useful for scientists working in medicinal chemistry, organic synthesis, who try to prepare these scaffolds.

I would like to mention that the quality of the work is certainly enhanced by the fact that in several cases the target compounds were produced even on a gram scale.

The following items should be addressed:

1.    In some cases, the characterization of the novel compound was described in detail which break the message of the article. Maybe it is better if these parts will moved to SI.

2.    Row 51. “t-butyl ( N-benzylazetidine-3-yl) carboxylate” instead of “ (N-benzyl)-1-azetidinyl t-butyl ester”

3.    In row 88-89: “gem-dimethyl bioisosteres, due to their similar dipole moments and ability to form hydrogen bonds,” Please double check this part of the sentence, methyl group is an apolar moiety.

4.    In methods section, please insert the retention factor (r.f.) of the products using TLC.

5.    HPLC analysis was mentioned in row 428, however, LC/MS was mentioned in row 236. Is it the same? Please add the column, eluent also to the description.

6.    Data Availability Statement: The data presented in this study are available on request from the corresponding authors. However, nowadays it can be useful if the data like fid files in NMR, etc. were uploaded to a database like Mendeley data.

7.    SI: Please insert the full spectra, from 0 to 10 ppm in the case of 1H NMR, 0 to 200 ppm in the case of 13C NMR.

Round 2

Reviewer 1 Report

In my point of view the manuscript can be accepted for publication in the present form.